# Diet modulates the therapeutic effects of dimethyl fumarate mediated by the immunometabolic neutrophil receptor HCAR2

Joanna Kosinska[1†], Julian C Assmann[1†], Julica Inderhees[1,2], Helge Müller-Fielitz[1], Kristian Händler[3], Sven Geisler[4], Axel Künstner[5], Hauke Busch[5], Anna Worthmann[6], Joerg Heeren[6], Christian D Sadik[7], Matthias Gunzer[8,9], Vincent Prévot[10], Ruben Nogueiras[11], Misa Hirose[5], Malte Spielmann[3], Stefan Offermanns[12], Nina Wettschureck[12], Markus Schwaninger[1]*

[1]Institute for Experimental and Clinical Pharmacology and Toxicology, Center of Brain, Behavior and Metabolism, University of Lübeck, Lübeck, Germany; [2]Bioanalytic Core Facility, Center of Brain, Behavior and Metabolism, University of Lübeck, Lübeck, Germany; [3]Institute of Human Genetics, Universitätsklinikum Schleswig-Holstein, University of Lübeck and University of Kiel, Lübeck, Germany; [4]Cell Analysis Core Facility, University of Lübeck, Lübeck, Germany; [5]Lübeck Institute of Experimental Dermatology, University of Lübeck, Lübeck, Germany; [6]Department of Biochemistry and Molecular Cell Biology, Center for Experimental Medicine, University Medical Center Hamburg-Eppendorf, Hamburg, Germany; [7]Department of Dermatology, Allergy, and Venereology, University of Lübeck, Lübeck, Germany; [8]Institute for Experimental Immunology and Imaging, University Hospital, University of Duisburg-Essen, Essen, Germany; [9]Leibniz-Institute for Analytical Sciences-ISAS, Dortmund, Germany; [10]Univ. Lille, Inserm, CHU Lille, Laboratory of Development and Plasticity of the Neuroendocrine Brain, Lille Neuroscience & Cognition, Lille, France; [11]Universidade de Santiago de Compostela-Instituto de Investigation Sanitaria, Santiago de Compostela, Spain; [12]Max Planck Institute for Heart and Lung Research, Department of Pharmacology, Bad Nauheim, Germany

*For correspondence:
markus.schwaninger@uni-luebeck.de

†These authors contributed equally to this work

**Competing interest:** The authors declare that no competing interests exist.

**Abstract** Monomethyl fumarate (MMF) and its prodrug dimethyl fumarate (DMF) are currently the most widely used agents for the treatment of multiple sclerosis (MS). However, not all patients benefit from DMF. We hypothesized that the variable response of patients may be due to their diet. In support of this hypothesis, mice subjected to experimental autoimmune encephalomyelitis (EAE), a model of MS, did not benefit from DMF treatment when fed a lauric acid (LA)-rich diet. Mice on normal chow (NC) diet, in contrast, and even more so mice on high-fiber (HFb) diet showed the expected protective DMF effect. DMF lacked efficacy in the LA diet-fed group despite similar resorption and preserved effects on plasma lipids. When mice were fed the permissive HFb diet, the protective effect of DMF treatment depended on hydroxycarboxylic receptor 2 (HCAR2), which is highly expressed in neutrophil granulocytes. Indeed, deletion of *Hcar2* in neutrophils abrogated DMF protective effects in EAE. Diet had a profound effect on the transcriptional profile of neutrophils and modulated their response to MMF. In summary, DMF required HCAR2 on neutrophils as well as permissive dietary effects for its therapeutic action. Translating the dietary intervention into the clinic may improve MS therapy.

## Editor's evaluation

In this valuable study the authors provide convincing evidence that dimethyl fumarate treatment of experimental autoimmune encephalomyelitis in mice required neutrophil HCAR2 as well as permissive dietary effects for its therapeutic action. This work reveals new insights into the role of dietary factors in the responsiveness to a disease modifying drug used widely in multiple sclerosis.

## Introduction

Orally available disease-modifying agents have made significant progress in the treatment of multiple sclerosis (MS). One of the most commonly prescribed drugs is dimethyl fumarate (DMF) (*Kern and Cepeda, 2020*). DMF and the recently approved diroximel fumarate are prodrugs of monomethyl fumarate (MMF) (*Hoogendoorn et al., 2021*). Besides MS, MMF or its prodrugs are also marketed for psoriasis and investigated for other conditions (*Hoogendoorn et al., 2021*). Randomized clinical trials and post-marketing studies have proven the efficacy of MMF or its prodrugs in relapsing-remitting MS (*Barros et al., 2020*; *Fox et al., 2012*; *Gold et al., 2012*; *Mallucci et al., 2018*; *Miclea et al., 2016*; *Zecca et al., 2020*). While DMF has a favorable safety profile compared to intravenous disease-modifying therapies, not all patients benefit (*Ontaneda et al., 2019*). In phase-III trials of DMF treatment, 64.9% of patients showed no signs of disease activity after 2 years (*Havrdova et al., 2017*), but conversely, about one-third of patients did not completely respond to DMF. Therefore, understanding the factors that determine the response to DMF is imperative and may help improve MS therapy. A polymorphism in the NOX3 gene that generates reactive oxygen species predicts DMF response (*Carlström et al., 2019*). Another study found that a rare population of memory T helper cells expressing high levels of neuroinflammatory cytokines and CXCR3 reflects DMF response in MS patients (*Diebold et al., 2022a*). Although these studies have revealed interesting mechanisms, their practical consequences for MS therapy remain unclear.

MMF, the active metabolite of DMF, activates the antioxidant transcription factor NRF2 by covalently binding to the regulatory factor KEAP1 (*Linker et al., 2011*). Subsequently, NRF2 inhibits the transcription of inflammatory genes (*Kobayashi et al., 2016*), a process that may contribute to the therapeutic effects of MMF. In addition, MMF is an agonist of the G protein-coupled receptor HCAR2 (GPR109A) in neutrophils, monocytes/macrophages, keratinocytes, and adipocytes (*Tang et al., 2008*). Notably, HCAR2 is required for the therapeutic efficacy of DMF in experimental autoimmune encephalomyelitis (EAE), a mouse model of MS, and in two mouse models of inflammatory skin diseases (*Chen et al., 2014*; *Suhrkamp et al., 2022*; *Wannick et al., 2018*). In line with *Hcar2* expression in myeloid cells, DMF treatment reduced neutrophil extravasation into CNS or skin (*Chen et al., 2014*; *Wannick et al., 2018*). A growing body of evidence supports a key role for neutrophils in MS. Myelopoiesis in the bone marrow is stimulated, and circulating neutrophils show signs of activation in MS patients (*Rumble et al., 2015*; *Shi et al., 2022*). Moreover, cell-specific targeting of neutrophils improved pathology in EAE models (*Khaw et al., 2020*; *Shi et al., 2021*). These data suggest the hypothesis that HCAR2 on neutrophils mediates DMF effects in EAE, but direct evidence is missing.

Initially, HCAR2 was identified as a receptor of nicotinic acid, a lipid-lowering drug (*Tunaru et al., 2003*). Via HCAR2, nicotinic acid reduces serum levels of free fatty acids (*Tunaru et al., 2003*). DMF treatment in MS patients has similar effects on serum lipids as nicotinic acid (*Bhargava et al., 2019*), supporting the notion that DMF activates HCAR2. In the meantime, further evidence has accumulated showing that HCAR2 is a key regulator of metabolism. An endogenous agonist of HCAR2 is the fasting metabolite β-hydroxybutyrate (*Offermanns, 2017*; *Taggart et al., 2005*). Moreover, HCAR2 activation protects against diet-induced obesity, and the metabolic effects of HCAR2 activation depend on diet (*Sato et al., 2020*; *Ye et al., 2020*; *Ye et al., 2019*). Therefore, we hypothesized that diet may influence therapeutic DMF effects also in MS, providing a possible explanation for why DMF does not benefit all patients. To test this hypothesis, we modified the dietary supply of fatty acids to mice in EAE, as medium-chain and short-chain fatty acids (SCFAs) have been shown to influence the course of disease (*Haghikia et al., 2015*). DMF treatment improved the neurological deficit in EAE if mice were fed a high-fiber (HFb) diet that leads to intestinal SCFA production but not a diet rich in lauric acid (LA), a medium-chain fatty acid. The DMF effect on HFb diet was lost in mice with a cell-specific deletion of *Hcar2* in neutrophils, suggesting that the diet may modulate neutrophil responses to DMF treatment. Indeed, diet had a marked effect on the transcriptional profile of neutrophils, and MMF

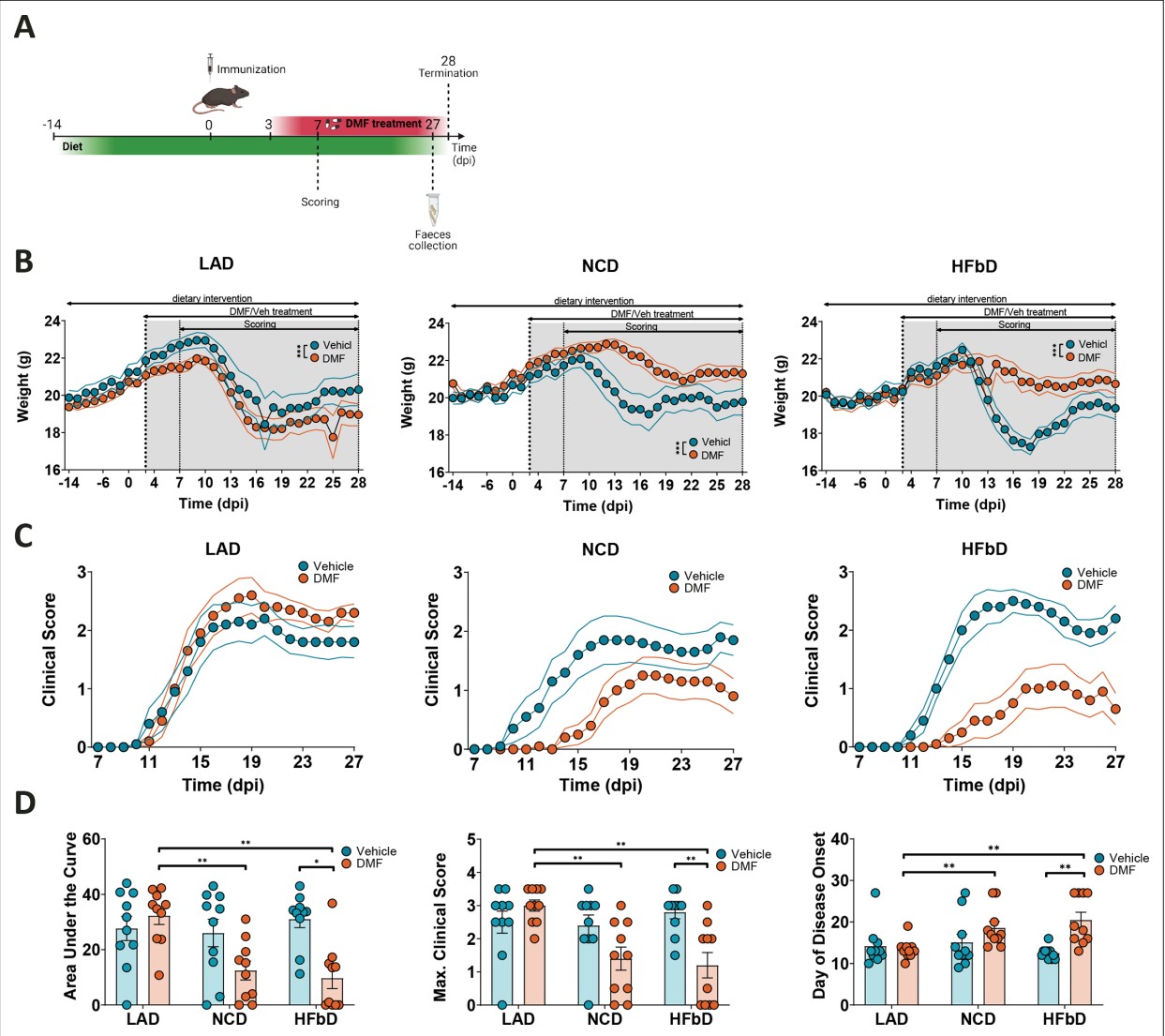

**Figure 1.** Diet modulates dimethyl fumarate (DMF) efficacy in experimental autoimmune encephalomyelitis (EAE). (**A**) Diet intervention started 14 days before immunization and was followed by DMF treatment (from dpi 3) and scoring (from dpi 7), feces collection 1 day before termination (dpi 28). (**B**) Body weight of mice on lauric acid diet (LAD), normal chow diet (NCD), and high-fiber diet (HFbD) during the course of EAE. DMF reduced weight loss in mice on NCD and HFbD, while the effect was lost in the LAD group. (**C**) Clinical scores in mice treated orally with vehicle or DMF (50 mg/kg body weight, twice daily). (**D**) Area under the curve of clinical scores, maximum scores, and the day of disease onset in mice on the three diets with or without DMF treatment. Dpi, day post-immunization. *p<0.05, **p<0.01, ***p<0.001. Means ± s.e.m. are shown. Points represent individual mice (**D**). Detailed information on the exact test statistics, sidedness, and values is provided in ***Supplementary file 6***.

## Results

### Diet modulates DMF efficacy in EAE

To test the impact of diets on the protective effects of DMF in EAE, we fed mice with LA diet, normal chow (NC) diet, or HFb diet starting 14 days before immunization (***Figure 1A***, ***Supplementary file 1***). At the onset of neurological symptoms 8–10 days after immunization, a decrease in body weight was observed (***Figure 1B***). Oral DMF treatment (50 mg/kg, twice per day) starting 3 days after immunization did not prevent weight loss in LA-treated mice, but had a moderate effect in NC-fed animals

inhibited neutrophil adhesion to endothelial cells and the formation of neutrophil extracellular traps (NETs) if mice were on HFb diet but less so on LA diet. Overall, the data demonstrate that diet determines the response of neutrophils to MMF and its therapeutic efficacy in experimental MS.

and a marked effect in HFb-fed mice (*Figure 1B*). In parallel, DMF's effect on the neurological deficit, which was assessed using a clinical scale, also depended on diet (*Figure 1C and D*). On LA diet, DMF had no effect, while there was a trend toward better clinical scores on NC diet and a clear improvement of the neurological deficit in terms of area under the curve (AUC), maximal score, and day of onset when mice received DMF treatment on HFb diet.

## Effects of diet and DMF on lipid metabolism

The interaction between treatment response and diet suggests that the resorption of orally administered DMF could be affected by diet. Therefore, we measured MMF in plasma 20 min after applying DMF by gavage to mice on the three diets. Importantly, diet had no significant effect on MMF plasma concentrations (*Figure 2A*). In accordance with clinical guidelines that DMF may be taken with or without food (Dimethyl Fumarate: MedlinePlus Drug Information), this indicates that the lack of DMF efficacy in LA diet-fed mice was not due to reduced resorption.

As expected, diet had a strong impact on plasma lipids. When we performed LC-MS/MS-based lipidomics on plasma samples obtained on day post-immunization (dpi) 28 in EAE and analyzed the data by sparse partial least-squares discriminant analysis (sPLS-DA), it became evident that key diet-induced differences (components 1 and 2) consisted of higher levels of medium-chain lipids and lower plasma levels of non-saturated lipids on LA diet than on HFb or NC diet (*Figure 2B*). Indeed, plotting the relative change of lipids on LA or HFb diet versus the number of double bonds demonstrated that saturated lipids preferentially increased on the LA diet while non-saturated lipids decreased (*Figure 2C*, *Supplementary file 2*). In contrast, mice on the HFb diet were characterized by higher levels of phospholipids containing non-saturated fatty acid residues (*Figure 2B*, component 2). As an example of a non-saturated fatty acid, plasma concentrations of FAs 18:2 including linoleic acid, the precursor of arachidonic acid, increased in the order of LA, NC, and HFb diet, in parallel with the functional response to DMF treatment (*Figure 2D*). Notably, this reflects different concentrations of linoleic acid in the diets (*Supplementary file 1*). Plasma concentrations of the SCFA acetic acid or propionic acid did not change in DMF-treated groups or between diets (*Figure 2—figure supplement 1*), and butyrate concentrations were below the limit of quantification.

Although DMF did not improve the clinical EAE score in combination with LA diet, it had metabolic effects in mice fed this diet. DMF significantly changed plasma lipids (*Figure 2E*, *Supplementary file 3*). Particularly in animals fed LA diet, DMF lowered triglycerides, mimicking the HCAR2-mediated effect of nicotinic acid (*Tunaru et al., 2003*). In addition, DMF increased some sphingomyelins and phosphatidylcholines (*Figure 2E*). DMF effects on plasma lipids were more pronounced when mice were on LA diet than on NC or HFb diet, as also shown by the separation based on component 3 of sPLS-DA (*Figure 2B*) only for LA diet groups. Even when DMF did not have marked effects on the lipid profile in mice fed an HFb diet, it protected against EAE, indicating that the lipid-lowering effects are not responsible for the protection in EAE. In addition, this observation supports the drug monitoring data showing that LA diet did not interfere with DMF resorption (*Figure 2A*).

## Effects of diet and DMF on small hydrophilic metabolites in plasma

To further characterize the interaction of DMF treatment and diet, we performed LC-MS/MS-based metabolomics of plasma samples obtained on dpi 28 in EAE (*Figure 3*). Diet had a significant impact on small hydrophilic metabolites in plasma as shown by sPLS-DA of the metabolomics dataset (*Figure 3A*, components 1 and 2). Component 1 metabolites included glycitein, an isoflavone, and equol, an isoflavone metabolite produced by gut bacteria (*Figure 3A*). Glycitein and equol levels were elevated in NC diet-fed mice, indicating that this diet is enriched in isoflavones (*Figure 3B*). Isoflavones and equol are protective in EAE (*Jensen et al., 2021*), but DMF treatment did not significantly change their plasma levels. The diet also affected metabolites of component 2, including *N*-acetyltyrosine and sphingosine 1-phosphate (S1P, *Figure 3A and C*). Plasma levels of *N*-acetyltyrosine were higher in the HFb diet group than in the LA diet group, similar to the clinical response to DMF treatment (*Figure 1C*). It is interesting to note that *N*-acetyltyrosine stimulates the expression of *Keap1*, a molecular target of MMF (*Matsumura et al., 2020*), suggesting a mechanism for how HFb diet could enhance the response to DMF. In contrast to *N*-acetyltyrosine, plasma levels of S1P were decreased in animals fed the HFb diet compared to the LA diet (*Figure 3C*). Lowering of S1P levels by HFb diet may have a permissive effect on the therapeutic effect of DMF, as S1P levels were elevated in MS

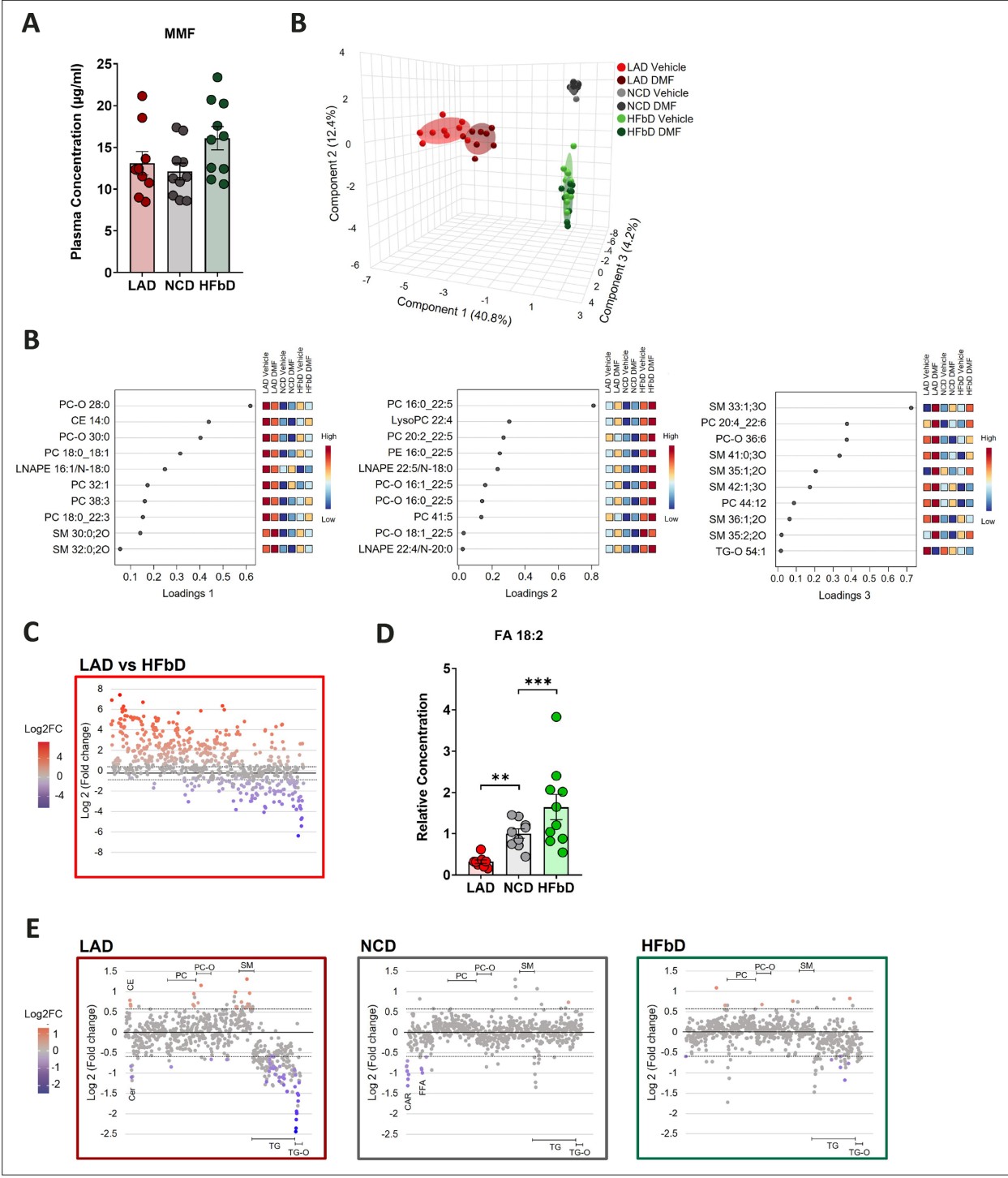

**Figure 2.** Effects of diet and dimethyl fumarate (DMF) on metabolism. (**A**) Monomethyl fumarate (MMF) plasma concentrations in mice that received lauric acid diet (LAD), normal chow diet (NCD), or high-fiber diet (HFbD) and oral DMF (50 mg/kg body weight) 20 min before blood sampling. (**B**) Sparse partial least-squares discriminant analysis (sPLS-DA) plot for the lipidomics dataset with corresponding loading plots for components 1, 2, and 3 (top 10). (**C**) Effects of LAD versus HFbD on plasma lipids in mice treated with vehicle are shown as fold change (FC). Each dot represents an individual lipid, sorted by the number of double bonds. Colored lipids were significantly changed by the diet (n=9–10 per group, Mann-Whitney U test, p≤0.05 (false discovery rate [FDR] adjusted), FC threshold = 1.5 [dotted line]). (**D**) Relative plasma concentration of fatty acids (FA) 18:2 in mice treated with vehicle. (**E**) Effects of DMF treatment on plasma lipids within diet groups are shown as FC. Each dot represents an individual lipid, sorted by the lipid class. Colored lipids were significantly changed by the DMF treatment (n=8–10 per group, Mann-Whitney U test, p≤0.05 [unadjusted], FC threshold = 1.5 [dotted line]). Major lipid classes are depicted. CAR, carnitines; CE, cholesterol ester; Cer, ceramides; FFA, free fatty acids; LNAPE, *N*-acyl

*Figure 2 continued on next page*

*Figure 2 continued*

lysophosphatidylethanolamine; PC, phosphatidylcholines; PC-O, ether-linked phosphatidylcholines; SM, sphingomyelins; TG, triglycerides; TG-O, ether-linked triglycerides. Samples in (**B–E**) were obtained on day post-immunization 28 in EAE in the experiment shown in *Figure 1*. **p<0.01, ***p<0.001. Means ± s.e.m. are shown. Points represent individual mice (**A, B, D**). Detailed information on the exact test statistics, sidedness, and values is provided in *Supplementary file 6*.

The online version of this article includes the following figure supplement(s) for figure 2:

**Figure supplement 1.** Plasma concentration of short-chain fatty acids (SCFAs).

**Figure supplement 2.** Chromatograms and fragmentation spectra.

patients, and functional inhibitors of S1P receptors, such as fingolimod, improve neurological deficits in MS (*McGinley and Cohen, 2021*; *Zahoor et al., 2022*).

Metabolites mainly affected by DMF treatment (component 3, *Figure 3A*) included glutamyl-glutamine that was elevated by DMF treatment, possibly reflecting NRF2 activation by DMF as NRF2 is known to increase glutathione, a substrate for glutamyl-glutamine synthesis (*Figure 3D*; *Hayes and Dinkova-Kostova, 2014*; *Li et al., 1999*). Irrespective of diet, DMF lowered uracil levels (*Figure 3D*). The underlying mechanism remains unclear, but the effect may be significant because elevated uracil concentrations occur in MS patients (*Lazzarino et al., 2017*). If the increase in uracil is relevant for neuroinflammation, normalization of uracil levels may contribute to the DMF effect. However, it seems questionable whether this and the other observed changes in metabolism explain differences in DMF efficacy between diets.

## Effects of diet and DMF on microbiota

Clinical studies have shown that MS is associated with changes in the gut microbiota (*Correale et al., 2022*). Interestingly, DMF treatment modulates the microbiota and partially reverses MS-related changes (*Diebold et al., 2022b*; *Ferri et al., 2023*; *Katz Sand et al., 2019*; *Storm-Larsen et al., 2019*). To explain diet-dependent therapeutic efficacy, we wondered whether DMF-induced microbiota effects depend on diet. Bacterial 16S rRNA sequencing of feces sampled on dpi 27 demonstrated diet-associated changes in microbiota at the level of phylum, genus (*Figures 1A and 4A*), and operational taxonomic unit (OTU). In accordance with others' observations (*Cantu-Jungles and Hamaker, 2023*), in our experiment, HFb diet reduced α-diversity, a measure of bacterial diversity within individual hosts, in comparison to LA and NC diets (*Figure 4B*). DMF did not affect α-diversity. Also, β-diversity analyses, a measure of dissimilarity between groups, showed significant differences between the diets, but no effects of DMF treatment (*Figure 4B*). In comparison to LA and NC diets, HFb diet increased butyrate-producing bacteria while lowering pro-inflammatory species (*Figure 4A*). Compared to LA or NC diets, HFb diet was associated with a higher abundance of the butyrate producers *Prevotella* and *Parabacteroides*, which may correct altered levels of these bacteria reported in MS patients (*Correale et al., 2022*; *Figure 4C*). DMF treatment did not significantly affect the dietary effects of HFb diet on *Prevotella* and *Parabacteroides* abundance. In contrast, DMF treatment was associated with higher abundance of *Acetatifactor* in HFb diet-fed mice (*Figure 4C*), but this species was associated with severe deficits in EAE (*Montgomery et al., 2020*). In summary, the microbiota analysis confirmed the effectiveness of the dietary intervention. However, it could not identify microbial changes that likely explain why DMF therapeutic efficacy depends on diet.

## HCAR2 in neutrophils mediates the DMF effect

A previous study showed that HCAR2 is required for DMF's beneficial effect in EAE, when mice were fed an NC diet (*Chen et al., 2014*). To test whether DMF's mode of action depends on diet, we repeated the experiment with animals on HFb diet. Two weeks after changing NC to an HFb diet, we induced EAE in $Hcar2^{-/-}$ mice and $Hcar2^{+/+}$ littermate controls and treated the animals with vehicle or DMF. While DMF treatment improved the neurological deficit and reduced the AUC of clinical scores in $Hcar2^{+/+}$ mice, it was inactive in $Hcar2^{-/-}$ animals (*Figure 5A and B*). These findings confirm that HCAR2 mediates the DMF action, also on HFb diet.

In search of the *Hcar2*-positive cell population that mediates the DMF effect, we used $Hcar2^{mRFP}$ mice expressing the mRFP reporter under control of the *Hcar2* locus (*Hanson et al., 2010*). We investigated mRFP-positive cells in the blood and brain of mice fed with HFb or LA diet and treated with

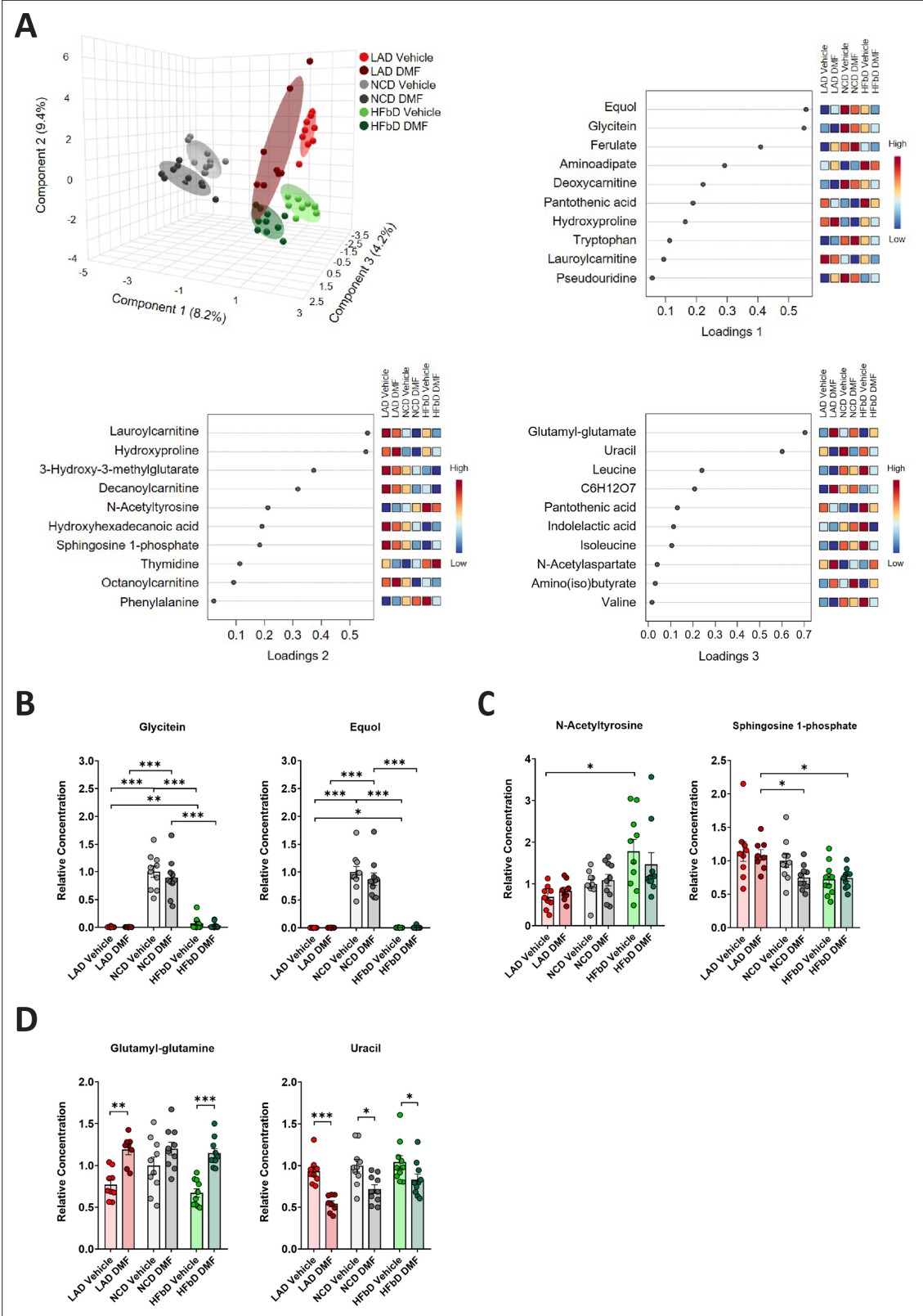

**Figure 3.** Effects of diet and dimethyl fumarate (DMF) on metabolic parameters. (**A**) Sparse partial least-squares discriminant analysis (sPLS-DA) plot for the metabolomics dataset with corresponding loading plots for components 1, 2, and 3 (top 10). Components 1 and 2 mainly reflected diet-induced effects, while component 3 reflected DMF-induced effects. (**B**) Relative plasma concentrations of the isoflavone glycitein and the isoflavone metabolite equol in mice treated with different diets and with or without DMF. (**C**) Relative plasma concentrations of *N*-acetyltyrosine and sphingosine 1-phosphate

*Figure 3 continued on next page*

Figure 3 continued

in mice treated with different diets and with or without DMF. (D) Relative plasma concentrations of glutamyl-glutamine and uracil in mice treated with different diets and with or without DMF. Samples were obtained in the experiment shown in **Figure 1**. *p<0.05, **p<0.01, ***p<0.001. Means ± s.e.m. are shown. Points represent individual mice. Detailed information on the exact test statistics, sidedness, and values is provided in **Supplementary file 6**.

vehicle or DMF after inducing EAE. In blood, we could not detect the reporter in lymphocytes, but almost all neutrophils were mRFP-positive, expressing the reporter at a high level (**Figure 6A**). In contrast, only part of patrolling Ly6C$^{low}$ and Ly6C$^{int}$ monocytes expressed *mRFP* at an intermediate level, and almost all inflammatory Ly6C$^{hi}$ monocytes were mRFP-negative (**Figure 6B–D**). The rate of mRFP-positive cells and expression per cell did not differ between LA and HFb diets or DMF and vehicle treatment, with the exception that HFb diet slightly increased *mRFP* expression in Ly6C$^{int}$ monocytes (**Figure 6C**). The upregulation of *Hcar2* expression in Ly6C$^{int}$ monocytes on HFb diet may contribute to mice's enhanced sensitivity to DMF treatment. However, independent of diet and DMF treatment, the highest expression of *mRFP* in blood cells was found in neutrophils, consistent with high levels of HCAR2 in neutrophils (**Tang et al., 2008**). In addition to neutrophils, microglia and monocyte-derived macrophages in the brain expressed *mRFP,* with HFb diet slightly increasing the *mRFP* expression in microglia (**Figure 6—figure supplement 2**).

Because of their high *Hcar2* expression, we selected neutrophils to investigate HCAR2 function in a cell-specific manner. To this end, we crossed *Hcar2$^{Fl/Fl}$* mice with the neutrophil-specific Ly6G-Cre driver line Catchup (**Hasenberg et al., 2015**). To test recombination efficiency, we measured intracellular Ca$^{2+}$ concentrations ([Ca$^{2+}$]$_i$) in neutrophils (**Figure 7**). As shown before (**Kostylina et al., 2008**), nicotinic acid (100 µM) elevated [Ca$^{2+}$]$_i$ and MMF (100 µM) had a similar effect (**Figure 7A–C**). Stimulation with nicotinic acid was mediated by HCAR2 as it increased [Ca$^{2+}$]$_i$ in wild-type *Hcar2$^{+/+}$* neutrophils but had no effect on neutrophils from *Hcar2$^{-/-}$* mice (**Figure 7A and B**). As *Hcar2* is an intron-less gene, we had inserted an artificial intron in the gene in addition to the loxP sites for the conditional knockout approach (**Suhrkamp et al., 2022**). *Hcar2$^{+/+}$* and *Hcar2$^{Fl/Fl}$* neutrophils reacted similarly to nicotinic acid and MMF stimulation, both with respect to the response of individual neutrophils and the rate of responsive cells, indicating that the artificial intron does not interfere with *Hcar2* expression (**Figure 7A–D**). However, in the presence of the *Ly6g$^{Cre}$* gene, the response rate dropped significantly from 75–80% in *Hcar2$^{Fl/Fl}$* neutrophils to about 35–40% in neutrophils from *Hcar2$^{nKO}$* mice (**Figure 7B and D**). As previously reported for other genes, the incomplete loss of HCAR2 function may be due to the limited time of *Ly6g* expression in neutrophil life span (**Hasenberg et al., 2015**).

Since the Ly6G-Cre-mediated conditional approach successfully reduced neutrophil responses to HCAR2 stimulation, we used this mouse model for EAE experiments. In *Hcar2$^{Fl/Fl}$* animals fed on HFb diet, DMF treatment had a significant effect on the neurological deficit. Interestingly, this effect was lost in *Hcar2$^{nKO}$* mice (**Figure 8A and B**), demonstrating that HCAR2 stimulation of neutrophils is required for DMF therapeutic effects.

## Diet modulates neutrophil response to DMF treatment

Since the therapeutic DMF effect was enhanced by HFb diet and depended on neutrophils' HCAR2, we wondered whether diet could impact neutrophil function. We fed wild-type mice one of the three diets (LA diet, NC diet, HFb diet) for 2 weeks and isolated neutrophils from bone marrow. After treating neutrophils in vitro with vehicle or MMF, we performed bulk RNA sequencing. The diet had profound effects on neutrophils' transcriptional profile (**Figure 9A**). According to gene ontology (GO) enrichment analysis, neutrophils expressed genes associated with the terms 'regulation of cell activation', 'innate immune response', and 'negative regulation of immune system process' at a higher level in HFb diet than in LA diet (**Figure 9B**). MMF treatment had a more subtle effect on gene expression that was characterized by GO terms related to glutathione synthesis and anti-oxidant defense (**Figure 9B**), in line with previous reports that MMF is able to reduce respiratory burst and the formation of reactive oxygen species in neutrophils (**Hoffmann et al., 2018**; **Nibbering et al., 1993**). The MMF-regulated genes included *Gclc* and *Gclm*, two subunits of the glutamate-cysteine ligase that catalyzes the rate-limiting step of glutathione synthesis, as well as the glutathione synthase *Gss*. Interestingly, there was no difference between the groups in *Hcar2* expression, but *Btg2* and *Il1b* that have been shown to be upregulated in neutrophils of MS patients (**Shi et al., 2022**) were reduced by the MMF treatment or the diet, respectively (**Figure 9C**).

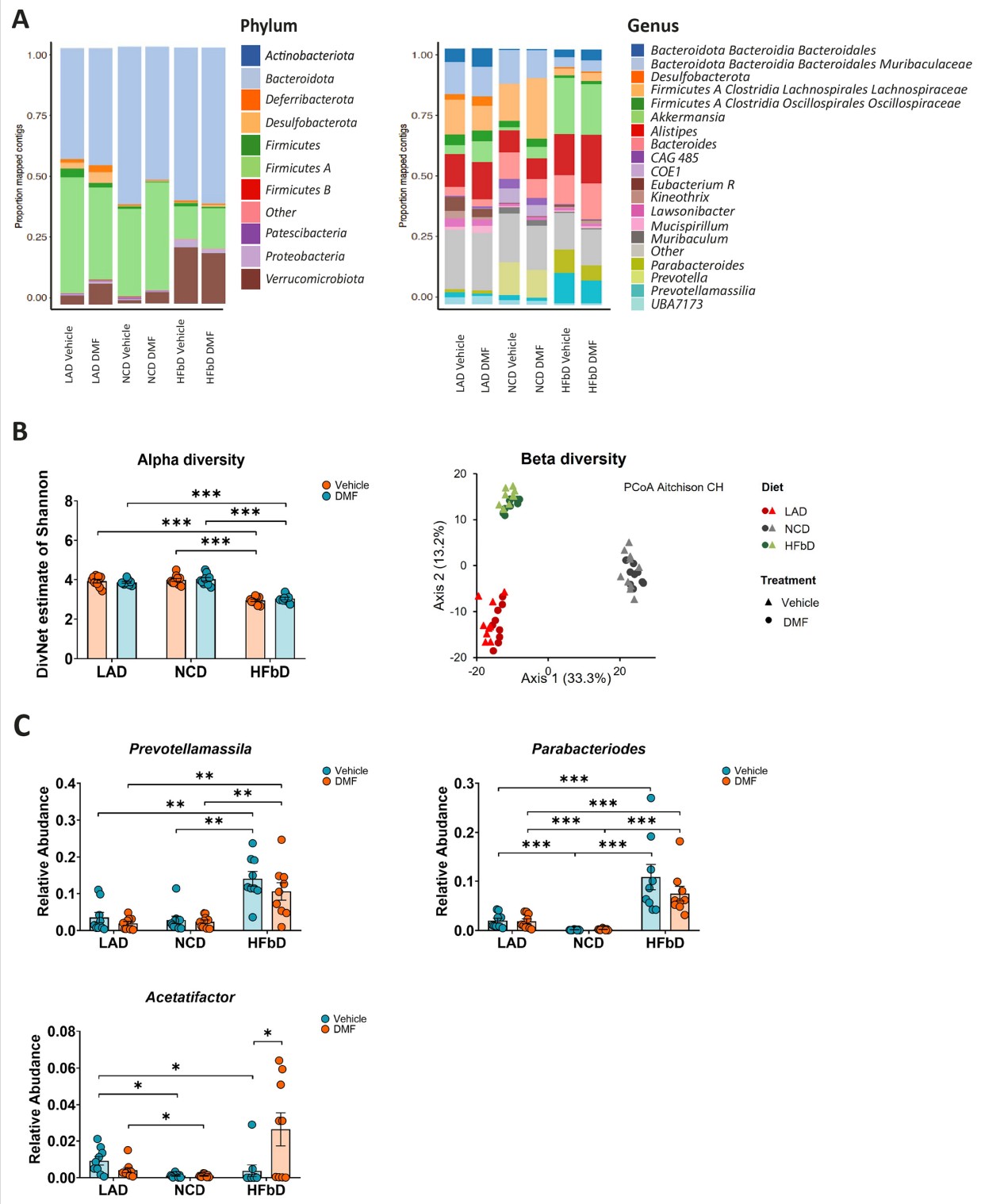

**Figure 4.** Effects of diet and dimethyl fumarate (DMF) on microbiota. (**A**) Diet affected the microbiota of mice subjected to experimental autoimmune encephalomyelitis (EAE) and treated with vehicle or DMF as indicated. High-fiber (HFbD) diet increased short-chain fatty acid (SCFA) producers at the phylum and genus level. (**B**) α-Diversity was significantly reduced by the HFb diet compared to lauric acid (LAD) and normal chow (NCD) diets, but DMF treatment had no effect. Similarly, only diet, not treatment, affected β-diversity as shown by principal coordinates analysis and PERMANOVA. (**C**) HFb diet or DMF treatment significantly increased the relative abundance of the SCFA producers *Prevotellamassilia* or *Parabacteroides* and *Acetatifactor*, respectively. Fecal samples were obtained from the experiment shown in *Figure 1* at day post-immunization (dpi) 28. *p<0.05, **p<0.01, ***p<0.001. Means ± s.e.m. are shown. Points represent individual mice. Detailed information on the exact test statistics, sidedness, and values is provided in *Supplementary file 6*.

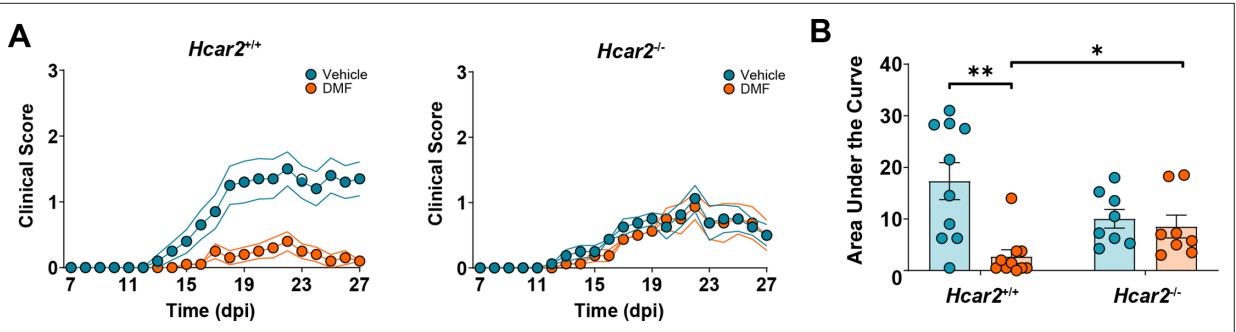

**Figure 5.** HCAR2 mediates the dimethyl fumarate (DMF) effect in experimental autoimmune encephalomyelitis (EAE) in mice on high-fiber (HFb) diet. Mice were switched to HFb diet 2 weeks before immunization and treated orally with vehicle or DMF (50 mg/kg body weight, twice daily) from day post-immunization (dpi) 3. (**A**) Clinical scores of *Hcar2⁺/⁺* and *Hcar2⁻/⁻* mice. DMF treatment significantly improved the neurological deficit in *Hcar2⁺/⁺* mice, while the effect was lost in *Hcar2⁻/⁻* mice, indicating that *HCAR2* mediates the protective effect of DMF. (**B**) Area under the curve of clinical scores in *Hcar2⁺/⁺* and *Hcar2⁻/⁻* mice treated with vehicle or DMF. *p<0.05, **p<0.01. Means ± s.e.m. are shown. Points represent individual mice (**B**). Detailed information on the exact test statistics, sidedness, and values is provided in ***Supplementary file 6***.

To evaluate the functional consequences of diet-induced neutrophil changes, we prepared primary neutrophils from mice fed with either LA, NC, or HFb diets for 4 weeks. Adherence of neutrophils to activated brain endothelial cells is a key step in their infiltration into the CNS. MMF significantly inhibited adhesion to activated bEnd.3 endothelial cells only when neutrophils were prepared from HFb diet-fed mice and not from LA or NC diet-fed animals (***Figure 10A***). Neutrophils perform immune functions in part by expelling DNA and other cellular components as NETs that are involved in MS pathology (***Woodberry et al., 2018***). Interestingly, MMF reduced NET formation only in neutrophils from HFb diet-fed mice, but no decrease was observed for animals on the other two diets (***Figure 10B***). The results of this study demonstrate that HFb diet in mice can permit HCAR2-mediated effects on neutrophils, which enhances their protective efficacy in EAE.

## Discussion

MMF and its prodrugs are a central pillar of MS therapy because of their oral bioavailability and their favorable safety profile. However, not all patients benefit from MMF or DMF treatment (***Havrdova et al., 2017***); therefore, this treatment strategy needs to be improved. Here, we report that diet critically determines the efficacy of DMF. DMF was most effective in EAE, the mouse model of MS, when animals were fed an HFb diet. Translating these findings into the clinic could improve MS therapy. To enable this process, we examined how dietary factors interact with DMF treatment.

Preclinical and clinical evidence indicates that oral administration of DMF leads to the activation of HCAR2, a G protein-coupled receptor activated by the DMF metabolite MMF (***Dubrall et al., 2021***; ***Hanson et al., 2012***; ***Tang et al., 2008***). In support of this, DMF mimicked the HCAR2-mediated effects of nicotinic acid on triglycerides in mice subjected to EAE and in human MS patients (***Figure 2***; ***Bhargava et al., 2019***; ***Tunaru et al., 2003***). Importantly, the therapeutic effects of DMF in EAE depended on HCAR2 (***Figure 5***; ***Chen et al., 2014***), because only wild-type but not HCAR2-deficient animals were protected by DMF treatment. Guided by the high expression of *Hcar2* in neutrophils, we deleted *Hcar2* in this cell type. In the absence of neutrophilic HCAR2, DMF did not have a protective effect in the EAE model, implicating neutrophils as a cellular target of DMF treatment (***Figure 8***). Whether *Hcar2*-expressing microglia or monocyte-derived macrophages contribute to the DMF action has to be determined in the future. Previous work had already indicated that DMF reduces neutrophil infiltration into the CNS during EAE (***Chen et al., 2014***). Although neutrophils have not been a traditional focus of research in MS or EAE, recent findings have revised this view (***Aubé et al., 2014***; ***Casserly et al., 2017***; ***Rumble et al., 2015***; ***Shi et al., 2021***). Neutrophils seem to be part of the early stages of the immune cascade that mediates neuroinflammation in EAE or MS (***Shi et al., 2022***). Importantly, they are linked to the adaptive immune system at multiple levels, potentially explaining how DMF treatment modulates dendritic cell and lymphocyte function (***Rosales, 2020***).

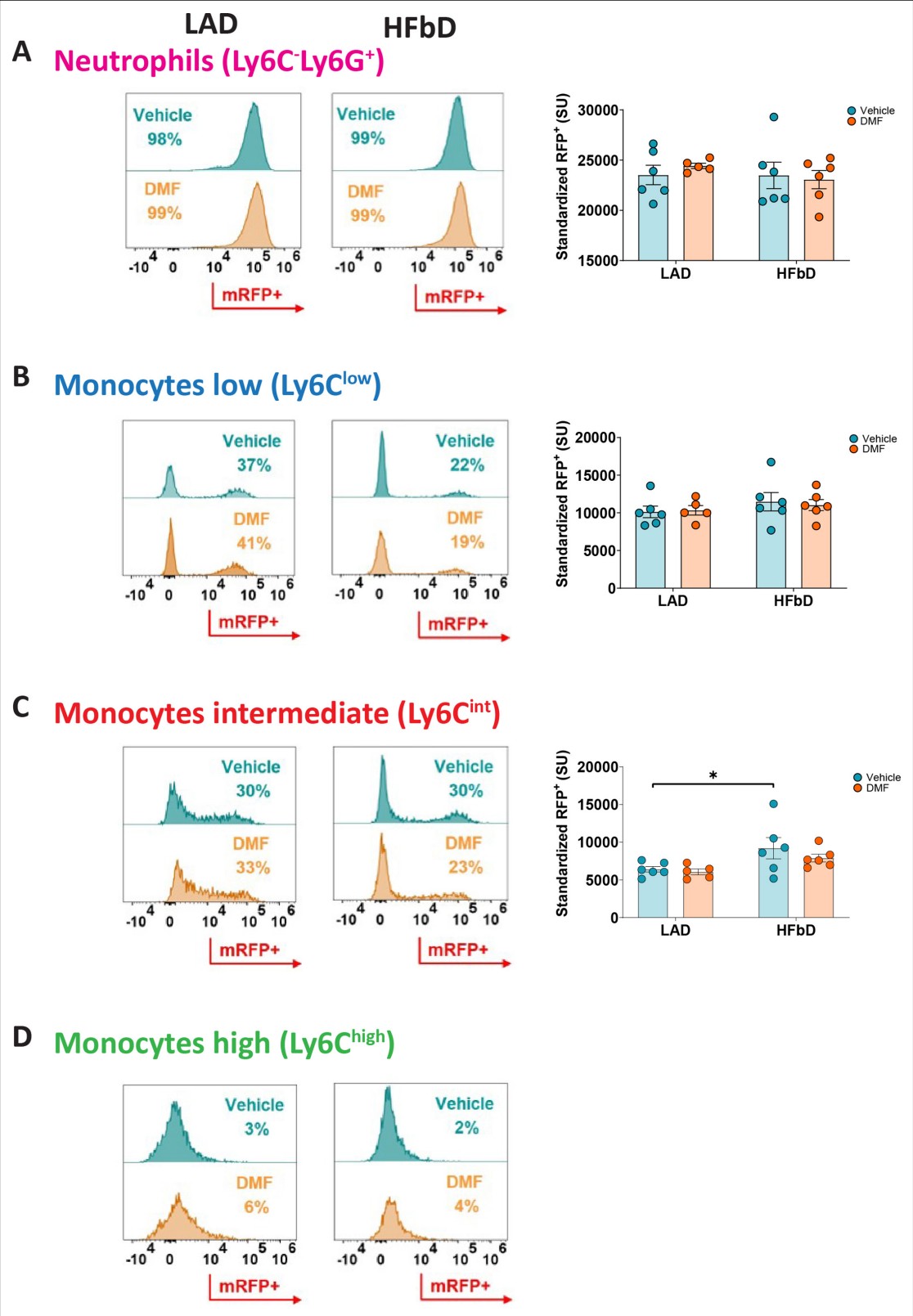

**Figure 6.** *Hcar2* reporter mice *Hcar2^mRFP* indicate that diet and dimethyl fumarate (DMF) treatment have no marked effects on *Hcar2* expression in peripheral immune cells of mice subjected to experimental autoimmune encephalomyelitis (EAE). After feeding *Hcar2^mRFP* mice with high-fiber diet (HFbD) or lauric acid diet (LAD) for 2 weeks, EAE was induced, and animals were treated orally with vehicle or DMF (50 mg/kg body weight, twice daily from day post-immunization [dpi] 3). On dpi 16/17, peripheral immune cells were analyzed by FACS. For the gating strategy, see **Figure 6—**

*Figure 6 continued on next page*

*Figure 6 continued*

**figure supplement 1.** The count of neutrophils or monocytes was not different between experimental groups (*Figure 6—figure supplement 1B*). (**A**) Histograms show that in both diets, regardless of treatment, more than 95% of neutrophils were RFP-positive. *Hcar2* expression per neutrophil, represented as standardized unit (SU) of mRFP$^+$, did not depend on DMF treatment or diet. (**B–D**) Among monocyte populations, Ly6C$^{int}$ and Ly6C$^{low}$ monocytes expressed mRFP at intermediate levels while almost all pro-inflammatory Ly6C$^{high}$ monocytes were mRFP-negative. DMF did not affect their *Hcar2-mRFP* expression per cell in both diets. HFbD slightly increased mRFP expression in Ly6C$^{int}$ monocytes. *$p<0.05$. Means ± s.e.m. are shown. Points represent individual mice (**B**). Detailed information on the exact test statistics, sidedness, and values is provided in *Supplementary file 6*.

The online version of this article includes the following figure supplement(s) for figure 6:

**Figure supplement 1.** Gating strategy of FACS measurement of mouse blood.

**Figure supplement 2.** *Hcar2* reporter mice *Hcar2*$^{mRFP}$ indicate that diet and dimethyl fumarate (DMF) treatment have no marked effects on *Hcar2* expression in immune cells in the brain of mice subjected to experimental autoimmune encephalomyelitis (EAE).

A central role for neutrophils in DMF action is supported by the observation that diet modulates the transcriptional profile of neutrophils and their response to MMF. On the HFb diet, neutrophils seem to be reprogrammed for several immune functions. Genes with GO terms related to cell activation were regulated. Although the expression of *Hcar2* was not altered by the diet, the response to MMF was stronger in mice fed an HFb diet, indicating a mode of action how the HFb diet may exert a permissive effect on DMF treatment. Of note, additional targets of DMF on cell types other than neutrophils have been previously described (*Kornberg et al., 2018*; *Linker et al., 2011*) and may enhance HCAR2-mediated neutrophil effects.

As diets are complex interventions, the identification of essential ingredients or actions is difficult. There are several ways in which diet might affect neutrophil function, however. Effects of HCAR2 in neutrophils may be blocked by the medium-chain fatty acid LA or saturated lipids derived from LA. A previous study reported that LA diet aggravates the untreated course of disease in EAE (*Haghikia et al., 2015*), but this effect was not evident in our experiments, perhaps because of a shorter time on this diet before immunization. Alternatively, SCFAs are candidates to mediate the effect of HFb diet because they are produced from dietary fibers and stimulate HCAR2, at least butyrate in high concentrations (*Taggart et al., 2005*). Since SCFAs were either unaltered or undetectable in the plasma of the HFb diet-fed mice, a systemic effect is unlikely, but we cannot exclude a local effect of butyrate in the gut wall. Indeed, HCAR2 mediates anti-inflammatory effects of HFb diet in the gut wall (*Offermanns, 2017*). Moreover, the immune system in the gastrointestinal tract, mainly intestinal T cells, has been linked to EAE (*Duc et al., 2019*; *Haupeltshofer et al., 2019*). Whether a local butyrate elevation in the gastrointestinal tract can modulate the effect of DMF treatment is unclear at present. Alternatively, dietary interventions had profound effects on other lipids and metabolites. Most conspicuous was the HFb diet-induced increase in non-saturated plasma lipids in DMF-responsive mice, when compared to LA diet-fed mice. Non-saturated lipids are substrates for the synthesis of immune-active components, such as resolvins that modulate neutrophil function (*Chiang et al., 2022*; *Li et al., 1996*). Finally, diet affects the gut microbiota, which in turn influences neutrophil function (*Lajqi et al., 2020*). In line with the activity of DMF as an NRF2 stimulus (*Linker et al., 2011*), we found evidence that DMF treatment has antioxidant effects in vivo and that MMF treatment stimulates an antioxidant transcriptional program in neutrophils.

Whatever the underlying mechanisms, translation into the clinic could improve the efficacy of MMF or its prodrugs in the treatment of MS or other diseases. The effect of MMF seems to be supported by a diet low in medium-chain saturated fatty acids, such as those found in coconut or palm oil, but rich in dietary fiber. Clinical trials are needed to verify this link.

## Materials and methods
### Animals

All mouse lines were established on a C57BL/6 background. Mice were 8–12 weeks of age. The sex of mice in individual experiments is specified in *Supplementary file 6*. Mice were housed in ventilated Green Line cages (Tecniplast) under controlled conditions (temperature 22 ± 1°C, light-dark cycle: 12:12 hr, lights on 7:00 am, relative humidity 55 ± 10%). Experimental procedures were approved by the local animal ethics committee (Ministerium für Landwirtschaft, ländliche Räume, Europa und Verbraucherschutz, Kiel, Germany, 106-8-17, 50-6-20, 28-4-22). For the investigation of plasma and

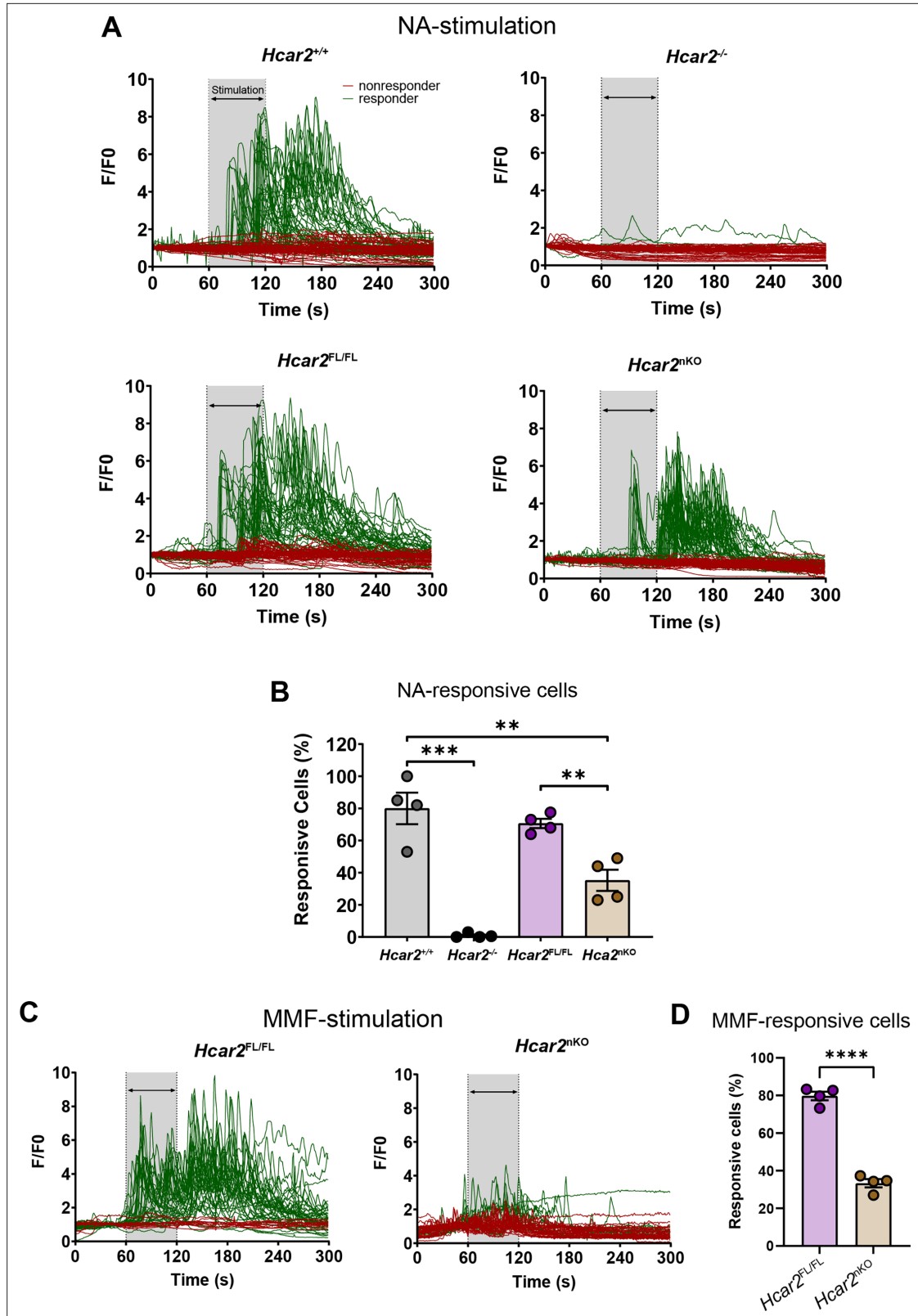

**Figure 7.** Conditional knockout of *Hcar2* in neutrophils (*Hcar2*nKO) decreases HCAR2-stimulated increase in intracellular $Ca^{2+}$ concentrations ([$Ca^{2+}$]i). Neutrophils were prepared from bone marrow of mice. (**A**) Stimulation with nicotinic acid (NA, 100 µM, 60 s) increased [$Ca^{2+}$]i concentrations in a majority of *Hcar2*+/+ neutrophils, while it had no effect in *Hcar2*-/- neutrophils. *Hcar2*Fl/Fl neutrophils responded in a similar manner to NA stimulation as *Hcar2*+/+ cells. The amplitude of [$Ca^{2+}$]i stimulated by nicotinic acid was lower in *Hcar2*nKO neutrophils. Representative traces of individual cells are

*Figure 7 continued on next page*

Figure 7 continued

shown. (**B**) The rate of neutrophils responding to NA (100 µM) was significantly reduced in *Hcar2nKO* mice compared to *Hcar2Fl/Fl* animals. (**C**) Stimulation with monomethyl fumarate (MMF) (100 µM) increased $[Ca^{2+}]_i$ in neutrophils from *Hcar2Fl/Fl* but not *Hcar2nKO* mice. (**D**) The rate of neutrophils responding to MMF (100 µM) was significantly reduced in *Hcar2nKO* mice compared to *Hcar2Fl/Fl* animals. **p<0.01, ***p<0.001, ****p<0.0001. Means ± s.e.m. are shown. Points represent individual mice. Detailed information on the exact test statistics, sidedness, and values is provided in *Supplementary file 6*.

microbiota during DMF therapy in EAE, as well as during the investigation of how diet modulates the sensitivity of neutrophils to MMF stimulation, C57BL/6N wild-type mice were ordered from Charles River Europe. Mice were fed with HFb diet (35% fiber, Cat. C1000 modified, #100213, Altromin), LA diet (30% fat, Cat. C1000 modified, #100212, Altromin), or NC (14% fat, 27% protein, 59% carbohydrates, Cat. #1314M, Altromin) ad libitum.

*Hcar2mRFP*, *Hcar2-/-*, *Hcar2Fl/Fl*, and *Ly6G-Cre* mouse lines were reported previously (*Hanson et al., 2010*; *Hasenberg et al., 2015*; *Suhrkamp et al., 2022*; *Tunaru et al., 2003*). As controls for *Hcar2-/-* mice, we used C57BL/6NCtrl wild-type mice ordered from Charles River Europe. For the conditional knockout in neutrophil granulocytes, we used *Ly6G-Cre; Hcar2Fl/Fl* (*Hcar2nKO*) mice and *Hcar2Fl/Fl* as controls.

## EAE induction and evaluation

EAE was induced by immunization with an emulsion of MOG$_{35-55}$ with complete Freund's adjuvant (CFA) (Hooke Kit MOG$_{35-55}$/CFA Emulsion PTX, EK-2110). The emulsion was injected subcutaneously at two sites in the upper and lower back (0.1 ml/site), followed by two intraperitoneal injections of pertussis toxin diluted with ice-cold PBS (100 ng in 0.1 ml/dose). Starting on dpi 7, mice were scored as follows: 0 – no disease, 0.5 – tip of tail is limp, 1 – limp tail, 1.5 – limp tail and hind limb inhibition, 2 – limp tail and weakness of hind limb, 2.5 – limp tail and dragging of hind limbs, 3 – limp tail and complete paralysis of hind limbs, 3.5 – limp tail, paralysis of hind limbs and weakness in forelimbs, 4 – humane endpoint. With the above immunization protocol, the prespecified humane endpoints (clinical score >3.5, loss of body weight >25%) were not reached in any mouse. DMF (Sigma-Aldrich) was prepared freshly every day and suspended in 0.8% Methocel (Sigma-Aldrich). Mice were given DMF (50 mg/kg body weight) or vehicle orally by gavage every 12 hr. The treatment was provided until the end of the study (dpi 28). In all experiments, we followed the ARRIVE guidelines. Investigators were blinded to the genotype of mice or to both genotype and treatment. Mice were randomly allocated to treatment groups.

## FACS

After inducing EAE, mice were deeply anesthetized via i.p. injection of ketamine/xylazine on dpi 16/17. Brains were homogenized, and cells were isolated by Percoll gradient centrifugation as described previously (*Chen et al., 2014*). When sampling blood from the right heart ventricle,

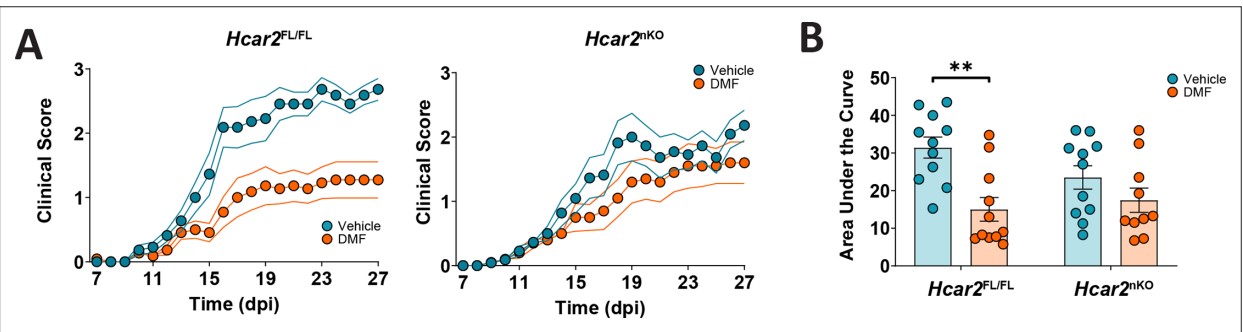

**Figure 8.** The therapeutic effect of dimethyl fumarate (DMF) in experimental autoimmune encephalomyelitis (EAE) depends on HCAR2 on neutrophils. (**A**) Clinical scores of *Hcar2FL/FL* and *Hcar2nKO* mice fed high-fiber (HFb) diet and treated orally with vehicle or DMF (50 mg/kg body weight, twice daily). DMF treatment significantly improved the neurological deficit in *Hcar2FL/FL* mice, while the effect was lost in *Hcar2nKO* mice, indicating that HCAR2 mediates the protective effect of DMF in neutrophils. (**B**) DMF treatment significantly reduced the area under the curve of clinical scores in *Hcar2FL/FL* but not in *Hcar2nKO* mice. **p<0.01. Means ± s.e.m. are shown. Points represent individual mice (**B**). Detailed information on the exact test statistics, sidedness, and values is provided in *Supplementary file 6*.

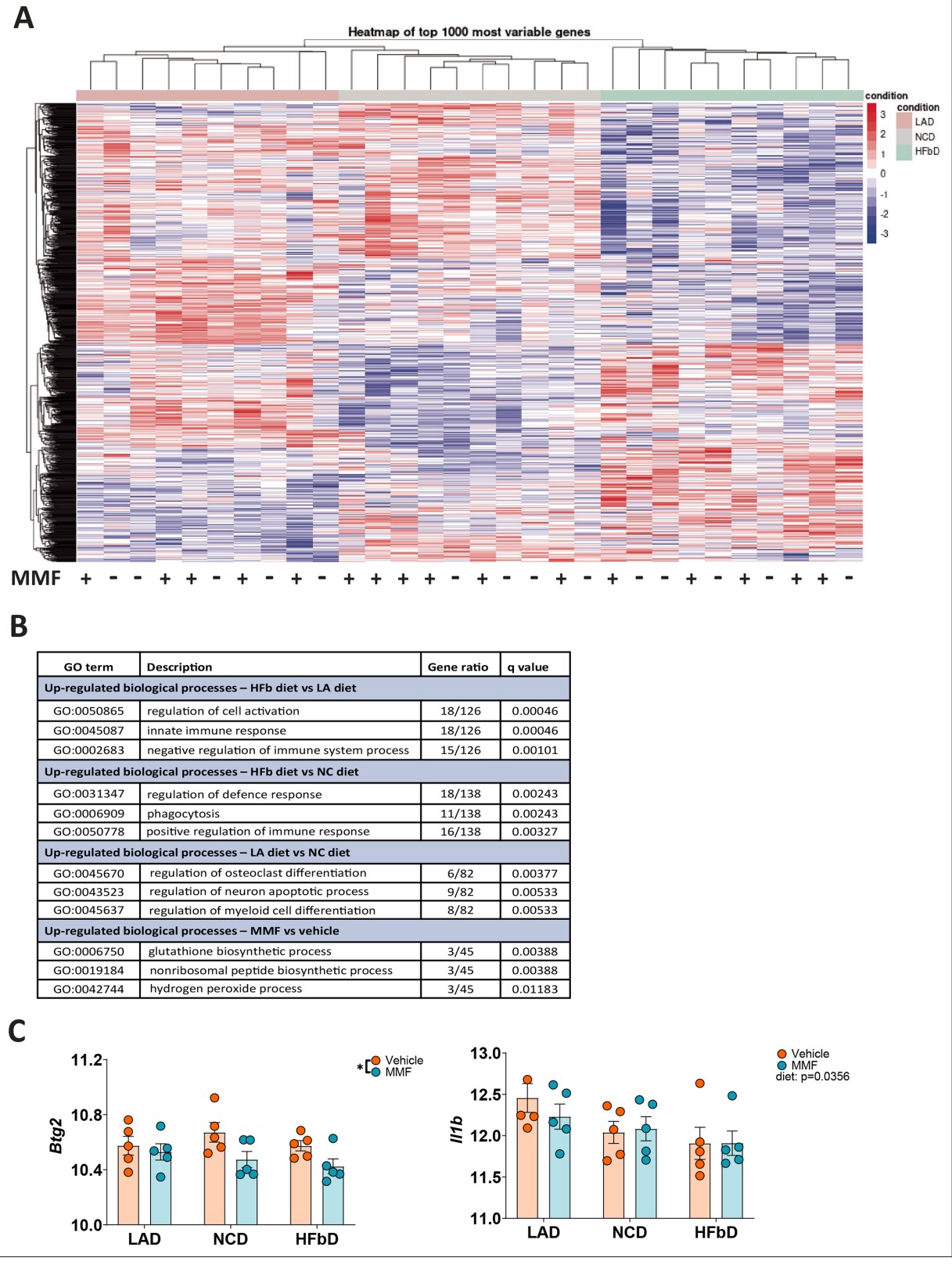

**Figure 9.** Dietary effects on the transcriptional profile of neutrophils. After mice were fed with lauric acid (LAD), normal chow (NCD), or high-fiber (HFbD) diets for 2 weeks, neutrophils were isolated from bone marrow and treated with vehicle or monomethyl fumarate (MMF, 100 μM, 3 hr) in vitro before bulk RNA sequencing. (**A**) Heatmap of the 1000 most variable genes is shown, indicating clear differences between diet groups. In contrast, MMF treatment of neutrophils had no marked effects. (**B**) Top ranking gene ontology (GO) terms identified by comparing diet or treatment groups. (**C**) Among

*Figure 9 continued on next page*

*Figure 9 continued*

five genes upregulated in neutrophils of MS patients (*Shi et al., 2022*), *Btg2* was downregulated by MMF treatment and *Il1b* by diet. *p<0.05. Means ± s.e.m. are shown. Points represent individual mice (**C**). Detailed information on the exact test statistics, sidedness, and values is provided in *Supplementary file 6*.

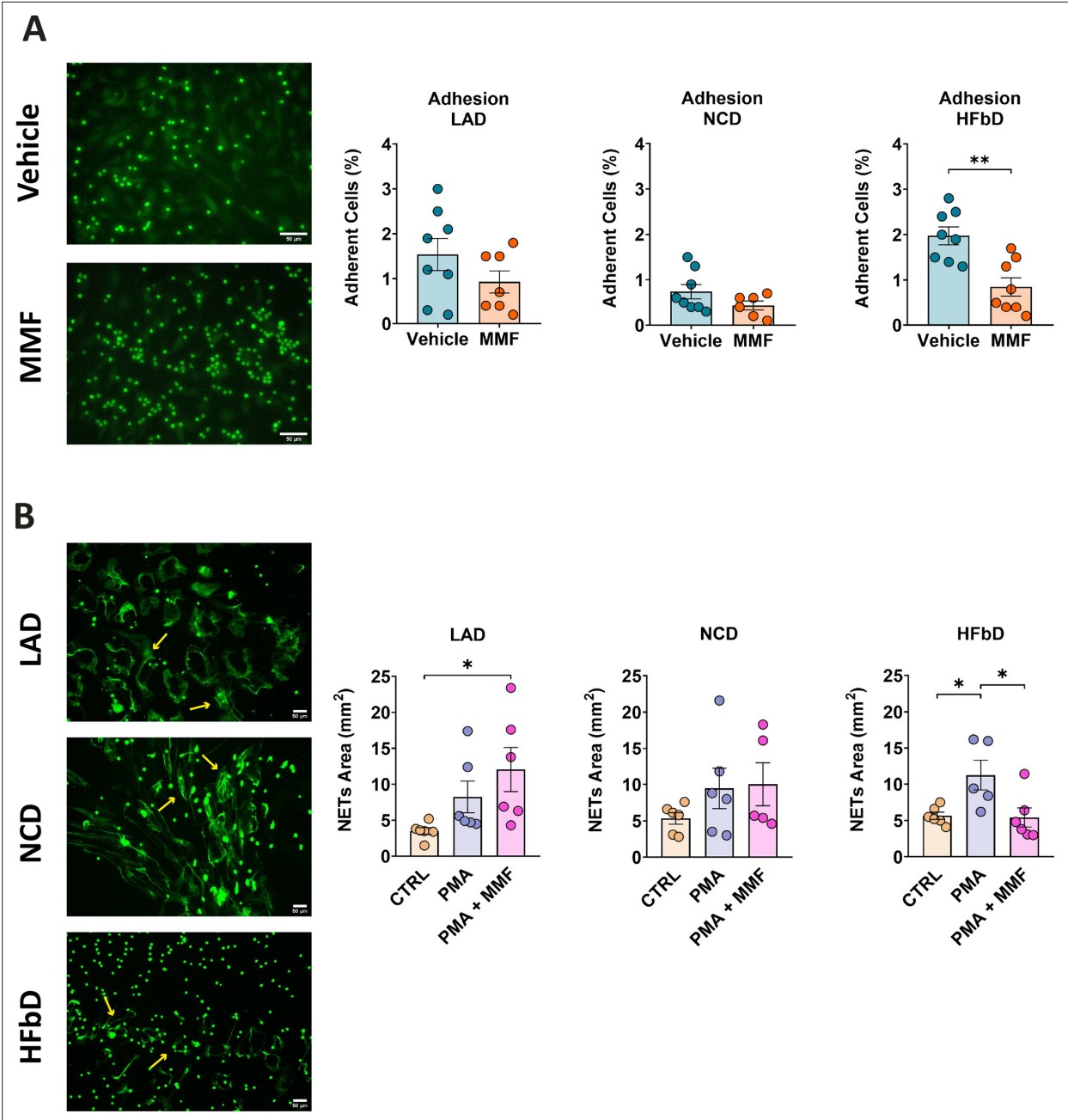

**Figure 10.** High-fiber diet (HFbD) has a permissive effect on the neutrophil response to monomethyl fumarate (MMF). Neutrophils were collected from the bone marrow of mice fed lauric acid diet (LAD), normal chow diet (NCD), or HFbD and stimulated with MMF (200 μM). (**A**) MMF significantly inhibited the adhesion of calcein-labeled neutrophils (green) to activated bEnd.3 brain endothelial cells when neutrophils were isolated from HFbD-fed mice, while no effects were observed in mice fed with LAD or NCD. In the left panels, representative images from the HFbD group are shown. (**B**) MMF treatment significantly reduced the phorbol 12-myristate 13-acetate (PMA)-triggered release of neutrophil extracellular traps (NETs, arrows) when mice were fed HFbD but not NCD or LAD. In the left panels, representative images of neutrophils treated with PMA + MMF of the three diets are shown. *p<0.05, **p<0.05. Means ± s.e.m. are shown. Points represent cultures from individual mice. Scale bars, 50 μm. Detailed information on the exact test statistics, sidedness, and values is provided in *Supplementary file 6*.

clotting was prevented by filling syringes with 200 µl 0.5 mM EDTA. Erythrocyte lysis buffer (5 ml, QIAGEN GmbH) was added to 1 ml EDTA blood, gently mixed, and incubated for 5 min at room temperature. To stop the lysis reaction, 10 ml PBS was added, followed by centrifugation. After the second lysis step, the supernatant was removed, and the cell pellet was resuspended in FACS buffer (300 µl, 0.5% bovine serum albumin [BSA] in PBS). To avoid nonspecific binding of immunoglobulins to Fc receptors, the cell suspension was incubated with Fc-block (Anti-Mouse CD16/CD32, 1:100, 4°C, in the dark) for 15 min. The reaction was stopped by adding FACS buffer (300 µl) and centrifuging the samples. After discarding the supernatant and resuspension, cells were incubated with a viability dye (eBioscience Fixable Viability Dye eFluor 780, 1:2000, 4°C, in the dark, 10 min). By adding FACS buffer (2 ml), incubation was stopped, and centrifugation was performed, followed by sample incubation with an antibody cocktail (300 µl, 20 min, 4°C, and in the dark, see *Supplementary file 5*). After cells were washed with FACS buffer (300 µl), centrifuged and resuspended in FACS buffer (400 µl), samples were ready to be measured using the spectral flow cytometer Cytek Aurora (Cytek Biosciences). The analysis was performed using FlowJo 10.10.0 software (BD Biosciences).

## Microbial DNA isolation, library preparation, and 16S rRNA gene sequencing

Fecal samples were collected from mice at dpi 28 in the chronic phase of EAE. Bacterial DNA isolation from fecal samples was performed using a QIAGEN Power Soil DNA Isolation Kit (QIAGEN, Hildfeld, Germany) according to the manufacturer's instructions. These isolated bacterial DNA samples were processed for 16S rRNA gene library preparation.

Library preparation of the bacterial 16S rRNA gene was performed as described previously (*Thomsen et al., 2023*). The hypervariable V3-V4 regions of the bacterial 16S rRNA gene were amplified by PCR using uniquely barcoded primers flanking the V3 and V4 hypervariable regions (341F-806R) with fused MiSeq adapters and heterogeneity spacers in a 25 µl volume, consisting of 1 µl of DNA (prior diluted 1:10), 4 µl of each forward and reverse primer (final conc. 2 µM), 0.25 µl of Phusion Taq II DNA polymerase (2 U/µl), 0.5 µl dNTPs (10 mM each), and 5 µl of HF buffer. The PCR conditions were as follows: initial denaturation for 30 s at 98°C; 30 cycles of 9 s at 98°C, 20 s at 55°C, and 45 s at 72°C; and a final extension for 10 min at 72°C. The concentration of PCR products was estimated on 1.5% agarose gels using Quantum Capt v16.04 software (Vilber Lourmat Deutschland GmbH) with a DNA marker (GeneRuler 100 bp Plus DNA Ladder; Thermo Fisher Scientific GmbH, Dreieich, Germany) as internal standard for band intensity measurement. Subsequently, the PCR products were pooled into approximately equimolar subpools, as indicated by band intensity, followed by clean-up of the subpools using a GeneJet column purification kit (Thermo Fisher). The concentrations of subpools were quantified using a Qubit dsDNA BR Assay Kit on a Qubit fluorometer (Thermo Fisher Scientific GmbH). The subpools were combined into one equimolar final pool. The final library was cleaned up with Agencourt AMPure Beads (Beckman Coulter GmbH). The concentration of the final pool was determined by qPCR using an NEBNext Library Quant Kit for Illumina (New England BioLabs GmbH, Hessen, Germany), and the size of the final library was measured by Bioanalyzer 2100 using Agilent DNA High Sensitivity Kit (Agilent Technologies GmbH, Waldbronn, Germany). Ten pM of the final library was sequenced on the Illumina MiSeq platform using MiSeq Reagent Kits v3 chemistry, generating 2×300 bp paired-end reads (Illumina, San Diego, CA, USA).

Demultiplexed forward and backward reads were merged (minimum overlap 100 bp, 1 mismatch), and merged reads were quality filtered (maxee = 0.5) using vsearch (v2.12.0) (*Rognes et al., 2016*). Next, chimeras were removed from the data using RDP gold v9 as the reference database; the remaining sequences were clustered into OTUs. Taxonomic assignments were performed by applying the SINTAX algorithm against GTDB (r202) (*Parks et al., 2018*). Potential contaminations were removed from the data using a frequency and prevalence approach (decontam v1.10.0; threshold 0.3; 16 OTUs removed from the data) (*Davis et al., 2018*). α-Diversity (Shannon estimator) was estimated using DivNet (v0.3.6) (*Willis and Martin, 2022*), and β-diversity was estimated on *clr*-transformed abundances using Euclidean distances (Aitchison distance) (*Aitchison, 1986*). Differences in β-diversity were assessed using permutational multivariate analysis of variance using distance matrices (PERMANOVA; vegan v2.6-4) with 9999 permutations to calculate p values.

## MMF measurement via LC-MS/MS

On the day of blood collection, 20 µl of fresh sodium fluoride (250 mg/ml, NaF) in water was added to each EDTA-containing tube. After mice were deeply anesthetized, blood was sampled from the right heart ventricle 20 min after DMF gavage, and 500 µl were transferred to the prepared EDTA tube. After centrifugation (1500×*g*) at 4°C for 10 min, plasma was snap-frozen in liquid nitrogen and stored at –80°C until analysis.

Methanol (150 µl containing 0.1% formic acid, Biosolve, Dieuze, France) and MMF-d3 (1.67 µg/ml, Toronto Research Chemicals, Toronto, Canada) as internal standard were added to plasma (50 µl). After mixing (1000 rpm, 4°C, 15 min; ThermoMixer, Eppendorf) and centrifugation (20,817×*g*, 4°C, 10 min), the supernatant was dried under vacuum and resuspended in 50 µl of acetonitrile/water (5:95, vol/vol) containing 0.1% formic acid for LC-MS/MS analysis. A blank plasma sample spiked with 5 µg/ml MMF was also analyzed. Samples were measured in randomized order. All solvents were of LC-MS grade quality and purchased from Merck (Darmstadt, Germany).

LC-MS/MS analysis was performed on a TSQ Endura triple quadrupole mass spectrometer, equipped with a heated electrospray ionization source and coupled to a Dionex Ultimate 3000 UHPLC system (Thermo Fisher Scientific). For LC, we used a Cortecs T3 column (100 mm × 2.1 mm, 2.7 µM; Waters), water and acetonitrile (LC-MS grade, Merck, Darmstadt, Germany), both containing 0.1% formic acid as eluent A and B, respectively. We applied the following gradient with a flow rate of 0.2 ml/min: 5% B for 0.5 min, increasing B to 65% between 0.5 and 1.5 min and to 100% until 4 min, then a washing step with 100% B between 4 and 6 min and decrease to 5% B within 0.5 min. Re-equilibration at 5% B was applied for 2 min. The following parameters were used in the negative ionization mode: ion spray voltage, 3.5 kV; vaporizer temperature, 200°C. Sheath and auxiliary gas were 35 and 7 (arbitrary unit), respectively. Selected reaction monitoring was performed with collision-induced fragmentation (collision energy = 30 V) at 1.5 mTorr using argon. Quantification was based on the deuterium-labeled internal standard (MMF-d3). Following transitions were used: m/z 129.09 → 31.02 and m/z 129.09 → 85.03 for MMF and m/z 132.04 → 34.04 and m/z 132.04 → 88.05 for MMF-d3. As the ratio for the most intense fragment of MMF (m/z 31.02) to MMF-d3 (m/z 34.04) in the spiked plasma samples (5 µg/ml each) was <1, a correction factor was applied for quantification (*Figure 2—figure supplement 2*).

## Lipidomics and metabolomics analyses via LC-MS/MS

All experiments used a Dionex Ultimate 3000 UHPLC system coupled to a Q-Exactive Orbitrap mass spectrometer (Thermo Fisher Scientific) equipped with a heated electrospray ionization source. All solvents were of LC-MS grade quality and purchased from Merck (Darmstadt, Germany).

Lipid profiling was performed as described previously (*Aherrahrou et al., 2020*). Briefly, lipids were extracted by adding 1 ml of methanol/methyl tertiary-butyl ether/chloroform (1.33:1:1, vol/vol/vol), containing butylated hydroxytoluene (100 mg/l, Sigma-Aldrich, Schnelldorf, Germany) and 2.5 ml/l of SPLASH internal standard mix (Avanti Polar Lipids, Alabaster, AL, USA), to 50 µl of plasma. After incubation and centrifugation, supernatants were dried under vacuum and resuspended in 50 µl of methanol/isopropyl alcohol (1:1, vol/vol) for LC-MS/MS analysis. Extracted lipids were separated on an Accucore C30 RP column (150 mm×2.1 mm, 2.6 µm; Thermo Fisher Scientific) using acetonitrile/water 6:4 (vol/vol) as eluent A and isopropyl alcohol/acetonitrile (9:1, vol/vol) as eluent B. Ammonium acetate (10 mM, Sigma-Aldrich, Schnelldorf, Germany) and formic acid (0.1%, Biosolve, Dieuze, France) were added to both eluents. Ionization and data acquisition with data-dependent MS$^2$ scans (top 15) were performed as described previously (*Aherrahrou et al., 2020*; *Karsai et al., 2019*). Lipid species were identified according to exact mass, isotopic pattern, retention time, and fragmentation ions using Compound Discoverer 3.3 (Thermo Fisher Scientific) and two in silico databases (*Kind et al., 2013*). The area under the peak was normalized to the internal standard. Pooled samples at four concentrations and an extraction blank were used as quality controls and for quality control-based normalization.

Metabolic profiling was performed as described previously (*Folberth et al., 2020*). Briefly, 200 µl acetone/acetonitrile/methanol (1:1:1, vol/vol/vol), containing 2.5 µM Metabolomics Amino Acid Mix Standard (Cambridge Isotope Laboratories, Andover, MA, USA), were added to 50 µl plasma. After incubation and centrifugation, supernatants were dried under vacuum and reconstituted in 25 µl methanol/acetonitrile (1:1, vol/vol) for LC-MS/MS analysis. Metabolites were separated on a SeQuant ZIC-HILIC column (150 mm×2.1 mm, 5 µm; Merck) using water with 5 mM ammonium

acetate as eluent A and acetonitrile/eluent A (95:5, vol/vol) as eluent B. The gradient elution was set as follows: isocratic step of 100% B for 3 min, 100% B to 60% B in 15 min, held for 5 min, returned to initial conditions in 5 min and held for 5 min. Flow rate was 0.5 ml/min. Data acquisition with data-dependent $MS^2$ scans (top 10) was performed. Metabolites were identified based on exact mass, isotopic pattern, retention time, and fragmentation spectra using Compound Discoverer 3.3 (Thermo Fisher Scientific), an in-house library (*Folberth et al., 2020*), and the online library mzCloud. Pooled samples at four concentrations were used as quality controls and for quality control-based normalization.

In total, 657 lipids and 91 metabolites were identified with level-2 confidence (*Schymanski et al., 2014*) and passed the defined quality criteria, including linearity of the signal, no significant signal in the extraction blank, and low variability. The average relative standard deviation (coefficient of variation; CV%) for technical replicates (quality controls) over two batches was 4.7% for lipids and 3.1% for metabolites, respectively. A list of all lipids and metabolites with the corresponding CV% can be found in *Supplementary files 3 and 4*, respectively.

## Measurement of SCFAs

Measurement of SCFA was performed as described previously (*Rohde et al., 2022*). In brief, SCFAs were extracted from plasma (30 µl) using ethanol (293.75 µl), and deuterated internal standard mix (6.25 µl, d4-acetic acid, d6-propionic acid, and d7-butyric acid, 4 g/l each, Sigma-Aldrich) was added. After homogenization, precipitated proteins were removed by centrifugation (10 min, 13,000×*g*). After transferring the supernatant to a fresh tube, NaOH (0.8 M, 5 µl) was added, followed by evaporation of solvents via vacuum centrifugation. After adding EtOH (50 µl) and succinic acid (10 µl, 0.6 M), samples were analyzed by GC-MS (TRACE 1310 gas chromatograph/ISQ 7000 mass selective detector; ThermoFisher Scientific, Dreieich, Germany) equipped with a Nukol Fused Silica Capillary Column (30 m×0.25 mm×0.25 µm film thickness) (Supelco/Sigma-Aldrich, St. Louis, MO, USA). The injector, GC-MS transfer line, and ion source temperature were set to 200°C, 200°C, and 250°C, respectively. The flow rate of helium carrier gas initially started at 1.3 ml/min, was kept there for 0.2 min before dropping to 1.1 ml/min at a rate of 1 ml/min. Until minute 6.2, the flow rate was kept constant at 1.1 ml/min and was then ramped to 2.6 ml/min at a rate of 1 ml/min where it was kept for 5.1 min. Samples (1 µl) were introduced by splitless injection. The initial column temperature was set to 55°C and held for 1 min, followed by a ramp-up to 105°C at a rate of 11°C/min where it was held for 3 min. Finally, the column temperature was increased to 190°C at a rate of 30°C/min and kept at this temperature for 4 min. Total runtime was 16 min. The ionization was carried out in the electron impact mode at 70 eV. The analytes were quantified in the timed selected ion monitoring mode using the target ion and verified by confirmative ions. The instrument was operated, data were acquired, and analyzed using Chromeleon software. Quantification of SCFA was achieved by comparing internal standard normalized peak areas of analytes to the respective normalized peak areas from external calibration curves.

## Neutrophil isolation

Mice were anesthetized with isoflurane and sacrificed by decapitation. The bone marrow was flushed from the tibia and femur. To obtain a single-cell suspension, the bone marrow was passed through a 23-gauge needle, and samples were centrifuged for 10 min at 240×*g* and 4°C. Neutrophils were enriched by gradient centrifugation with Percoll or Histopaque. A Percoll (Cytiva, Merck) gradient with densities of 52%, 69%, and 78% was prepared, and neutrophils were separated using an ultracentrifuge (*Boxio et al., 2004*). The cells were collected from the 69/78% and 78% layers. For enrichment with Histopaque, a gradient of Histopaque 1077 and Histopaque 1119 was prepared, and the cell suspension was layered on top (*Swamydas and Lionakis, 2013*). Neutrophils were collected from the interface between Histopaque 1077 and Histopaque 1119 layers. Enriched neutrophils were purified from remaining erythrocytes by adding erythrocyte lysis buffer. After 2 min, incubation was stopped by adding HBSS buffer (10 ml). After centrifugation (10 min, 350×*g*, room temperature), the cell pellet was resuspended in HEPES buffer (1 ml) or RPMI medium. With both methods, neutrophil purity was >90%, as assessed by CytoSpin followed by Wright-Giemsa staining or FACS analysis.

## RNA-Seq

Neutrophils were isolated in five biological replicates from mice fed LA, NC, or HFb diets (5 mice per diet). Isolated neutrophils were incubated with either MMF (100 µM) or PBS as a control for 3 hr. Treated neutrophils (3–5 × $10^6$ cells per sample) were then pelleted and resuspended in TRIzol (1 ml, Invitrogen) before total RNA extraction via RNeasy Plus Universal Mini Kit (QIAGEN) following the manufacturer's protocol. RNA integrity was determined using an RNA 6000 Pico Kit assay on a Bioanalyzer 2100 device (Agilent Technologies). Libraries were generated using total RNA (5 µl per sample) and QuantSeq 3' mRNA-Seq V2 Library Prep Kit FWD with UDI (Lexogen) following the manufacturer's instructions. Final libraries were quantified via Qubit 1X dsDNA High Sensitivity (HS) Kit (Invitrogen), and library size distribution was determined with a High Sensitivity DNA Kit assay on a Bioanalyzer 2100 device (Agilent Technologies) to estimate library molarity. Libraries were pooled equimolarly and sequenced R1-100 I7-8 I5-8 with NextSeq 2000 P3 Reagents (100 Cycles) on a NextSeq 2000 device (Illumina). Sequencing data conversion and demultiplexing were performed using bcl2fastq2 v2.20 (Illumina). Gene-level count matrices were generated from the fastq files using the community-driven nf-core/rnaseq pipeline v3.12.0 (*Patel et al., 2024*). In short, the pipeline consisted of adapter trimming (TrimGalore! v0.6.7), contamination removal (SortMeRNA v4.3.4 and BBMap-BBSplit v39.01), alignment (STAR 2.7.10a) to mm10, transcript-level quantification (Salmon v1.10.1), and comprehensive quality control (FastQC v0.11.9, RESeQC v3.0.1, dupRadar v1.28.0, and MultiQC v1.14). The transcript-level counts were converted to gene-level counts using tximeta-tximport v1.12.0. Downstream differential expression analysis was carried out with Shiny-Seq (*Sundararajan et al., 2019*).

## Adhesion assay

Adhesion of mouse neutrophils to activated brain endothelial cells was investigated as described previously (*Chen et al., 2014*). Neutrophils were labeled with calcein-AM (Invitrogen, 3 µM, 30 min, 37°C) followed by treatment with MMF (200 µM, Sigma-Aldrich) or PBS as vehicle for 30 min at 37°C in the presence of carbenoxolone (100 µM) to inhibit calcein efflux through hemichannels (*Bargiotas et al., 2011*). After washing and resuspending neutrophils in phenol-free RPMI-1640 medium, we transferred them ($5×10^5$/well) to activated bEnd.3 mouse brain endothelial cells (ATCC, CRL-2299). bEnd.3 cells were cultured on 48-well plates in DMEM (FG0445, Merck) containing fetal calf serum (10%) and penicillin/streptomycin (100 U/ml, 0.1 mg/ml) at 37°C and 5% $CO_2$. For activation, confluent bEnd.3 cells were stimulated with TNF (10 ng/ml, 6 hr) and washed three times with 3% BSA in phenol-free RPMI-1640 medium. After incubating neutrophils on bEnd.3 cells for 20 min at 37°C, the cells were washed twice with PBS with $Mg^{2+}$ and $Ca^{2+}$ followed by 4% paraformaldehyde (PFA) for 15 min at room temperature.

## NET formation assay

After isolation from mouse bone marrow, neutrophils ($3×10^6$/ml) were primed with recombinant mouse TNF (315-01A, PeproTech; 2 ng/ml, 15 min, 37°C, in phenol-free RPMI-1640 medium). After changing the medium, cells were plated on 12 mm coverslips coated with fibronectin (10 µg/ml) in a 24-well plate. Cells were either only labeled with 5 µM Sytox Green (SG, Invitrogen), labeled with SG and stimulated with phorbol 12-myristate 13-acetate (PMA, 100 nM, Sigma-Aldrich), or labeled with SG and treated with PMA plus MMF (200 µM). After an incubation for 6 hr at 37°C, the supernatant was gently aspirated. Cells were fixed with 4% PFA for 15 min at room temperature followed by mounting coverslips on the slides. In images taken by fluorescence microscopy (Leica, DMI 6000B), NET formation was quantified by measuring the area of NETs using ImageJ.

## Imaging of intracellular $Ca^{2+}$

After isolation from mouse bone marrow, neutrophils ($2×10^5$/well) were plated on 12 mm fibronectin-coated coverslips in 24-well plates. Cells were incubated for 3 hr in HEPES buffer at 37°C. Then, the supernatant was gently removed, and cells were loaded with 2 µM Fluo-4 (Invitrogen) in artificial cerebrospinal fluid (aCSF, 130 mM NaCl, 26.5 mM $NaHCO_3$, 1.25 mM $NaH_2PO_4$, 3 mM KCl, 2 mM $CaCl_2$, 2 mM $MgCl_4$, 10 mM D-glucose, 0.5% DMSO, 0.05% Pluronic F-127 [Invitrogen]) for 30 min at 37°C. After aspirating the dye solution, we incubated neutrophils with 2.5 mM probenecid (Invitrogen) in aCSF for another 30 min at 37°C. To detect changes in the $[Ca^{2+}]_i$ response, neutrophils were placed in the flow chamber and measured using a high-speed calcium imaging setup (Till Photonics)

mounted on the Axio Examiner D1 upright fluorescent microscope (Zeiss) coupled to the polychrome V monochromator and a high-speed CCD camera (Retiga EXi-blue, QImaging). Data acquisition and quantification were done using life acquisition and offline analysis software. In the first minute of the experiment, cells were in the measurement buffer (aCSF, 5% $CO_2$, 95% $O_2$, 7.4 pH, and flow rate of 2 ml/min), and then the stimuli, 100 µM NA or 100 µM MMF, were applied for 60 s followed by recovery in the measurement buffer for another 3 min.

## Statistical analysis

Sample sizes were planned based on a power analysis by G*Power using the clinical score as the primary outcome parameter. Most data were analyzed using GraphPad Prism 8 (GraphPad Software), and significance was considered when $p < 0.05$. Depending on the dataset and experimental design, different statistical methods were used as indicated in *Supplementary file 6*. Parametric statistics (e.g. t-test, ANOVA) were only applied if assumptions were met, i.e., datasets were examined for Gaussian distribution by D'Agostino-Pearson test, aided by visual inspection of the data, and homogeneity of variances by Brown-Forsythe, Levene's, or F-test (depending on the statistical method used). If assumptions for parametric procedures were not met or could not be reliably assumed due to small sample size, non-parametric methods were used as indicated. Two-tailed tests were applied if not indicated otherwise. Fold change analysis and sPLS-DA of lipidomics and metabolomics data were performed using MetaboAnalyst 6.0 (*Xia et al., 2009*). For the comparison of dietary effects on lipidomics and metabolomics data, p values were adjusted for false discovery rate by the Benjamini-Hochberg procedure. All data are shown as means ± s.e.m. Details of the statistical analyses are presented in *Supplementary file 6*.

## Acknowledgements

We would like to thank Ines Stölting, Wiebke Brandt, and Nathalie Kruse, Lübeck, for expert technical assistance. The research leading to these results received funding from the Deutsche Forschungsgemeinschaft to MSc (SCHW 416/12-1) and NW (WE 2891/2-1) and from the European Research Council (ERC) under the European Union's Horizon 2020 research and innovation program (grant agreement No 810331) to VP, RN, and MSc. HB and AK acknowledge support through the BMBF (Outlive-CRC-01KD2103A).

## Additional information

### Funding

| Funder | Grant reference number | Author |
| --- | --- | --- |
| European Research Council | 10.3030/810331 | Vincent Prévot Ruben Nogueiras Markus Schwaninger |
| Deutsche Forschungsgemeinschaft | SCHW 416/12-1 | Markus Schwaninger |
| Deutsche Forschungsgemeinschaft | WE 2891/2-1 | Nina Wettschureck |
| Bundesministerium für Bildung und Forschung | Outlive-CRC-01KD2103A | Axel Künstner Hauke Busch |

The funders had no role in study design, data collection and interpretation, or the decision to submit the work for publication.

### Author contributions

Joanna Kosinska, Julica Inderhees, Data curation, Formal analysis, Validation, Investigation, Visualization, Methodology, Writing – original draft, Writing – review and editing; Julian C Assmann, Data curation, Formal analysis, Validation, Investigation, Methodology, Writing – review and editing; Helge Müller-Fielitz, Formal analysis, Investigation, Visualization, Methodology; Kristian Händler,

Investigation, Visualization, Writing – review and editing; Sven Geisler, Anna Worthmann, Joerg Heeren, Investigation, Methodology, Writing – review and editing; Axel Künstner, Software, Formal analysis, Visualization, Writing – review and editing; Hauke Busch, Software, Formal analysis, Writing – review and editing; Christian D Sadik, Matthias Gunzer, Resources, Writing – review and editing; Vincent Prévot, Ruben Nogueiras, Funding acquisition, Project administration, Writing – review and editing; Misa Hirose, Formal analysis, Supervision, Investigation, Writing – review and editing; Malte Spielmann, Formal analysis, Investigation, Writing – review and editing; Stefan Offermanns, Conceptualization, Resources, Writing – review and editing; Nina Wettschureck, Conceptualization, Resources, Project administration, Writing – review and editing; Markus Schwaninger, Conceptualization, Resources, Supervision, Funding acquisition, Writing – original draft, Project administration, Writing – review and editing

## Author ORCIDs

Joanna Kosinska https://orcid.org/0000-0003-3025-3685
Julica Inderhees https://orcid.org/0000-0003-4523-3652
Helge Müller-Fielitz https://orcid.org/0000-0003-2815-4426
Axel Künstner https://orcid.org/0000-0003-0692-2105
Joerg Heeren https://orcid.org/0000-0002-5647-1034
Matthias Gunzer https://orcid.org/0000-0002-5534-6055
Malte Spielmann https://orcid.org/0000-0002-0583-4683
Stefan Offermanns https://orcid.org/0000-0001-8676-6805
Markus Schwaninger https://orcid.org/0000-0002-4510-9718

## Ethics

Experimental procedures were approved by the local animal ethics committee (Ministerium für Landwirtschaft, ländliche Räume, Europa und Verbraucherschutz, Kiel, Germany, 106-8-17, 50-6-20, 28-4-22).

## Decision letter and Author response

Decision letter https://doi.org/10.7554/eLife.98970.sa1
Author response https://doi.org/10.7554/eLife.98970.sa2

# Additional files

## Supplementary files

Supplementary file 1. Main ingredients in the diets.

Supplementary file 2. Fold change of lipids on lauric acid (LAD) versus high-fiber (HFbD) diet and corresponding p values (Mann-Whitney U test, false discovery rate [FDR] adjusted), see Excel file.

Supplementary file 3. Fold change of lipids on dimethyl fumarate (DMF) versus vehicle treatment and corresponding p values (Mann-Whitney U test); list of quantified lipids with the CV%, see Excel file.

Supplementary file 4. List of quantified metabolites with the CV% of quality controls, see Excel file.

Supplementary file 5. Antibodies used for flow cytometry.

Supplementary file 6. Results of statistical analyses. f, female; m, male.

MDAR checklist

## Data availability

Raw RNA sequencing data have been deposited in Gene Expression Omnibus under accession number GSE263752. Lipidomics and metabolomics data are available at the NIH Common Fund's National Metabolomics Data Repository (NMDR) website, the Metabolomics Workbench, https://www.metabolomicsworkbench.org where it has been assigned Project ID (PR001962). The data can be accessed directly via https://doi.org/10.21228/M83X6D. This repository is supported by Metabolomics Workbench/National Metabolomics Data Repository (NMDR) (grant# U2C-DK119886), Common Fund Data Ecosystem (CFDE) (grant# 3OT2OD030544) and Metabolomics Consortium Coordinating Center (M3C) (grant# 1U2C-DK119889). All raw data not included in the article are available through GEO and the National Metabolomics Data Repository.

The following datasets were generated:

| Author(s) | Year | Dataset title | Dataset URL | Database and Identifier |
|---|---|---|---|---|
| Kosinska J, Assmann JC, Inderhees J, Händler K, Geisler S, Künstner A, Busch H, Worthmann A, Heeren J, Christian S, Matthias G, Prevot V, Nogueiras R, Hirose M, Spielmann M, Offermanns S, Wettschureck N, Schwaninger M, Müller-Fielitz H | 2024 | Diet modulates the protective effects of dimethyl fumarate mediated by the immunometabolic neutrophil receptor HCA2 | https://www.ncbi.nlm.nih.gov/geo/query/acc.cgi?acc=GSE263752 | NCBI Gene Expression Omnibus, GSE263752 |
| Inderhees J, Kosinska J, Assmann JC | 2024 | Diet modulates the protective effects of dimethyl fumarate mediated by the immunometabolic neutrophil receptor HCA2 | https://doi.org/10.21228/M83X6D | NIH Common Fund's National Metabolomics Data Repository, 10.21228/M83X6D |

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
