## [Editor Report]

In this valuable study the authors provide convincing evidence that dimethyl fumarate treatment of experimental autoimmune encephalomyelitis in mice required neutrophil HCAR2 as well as permissive dietary effects for its therapeutic action. This work reveals new insights into the role of dietary factors in the responsiveness to a disease modifying drug used widely in multiple sclerosis.

---

## [Decision Letter]

**Decision letter after peer review:**

Thank you for submitting your article "Diet modulates the protective effects of dimethyl fumarate mediated by the immunometabolic neutrophil receptor HCA_2_" for consideration by *eLife*. Your article has been reviewed by 2 peer reviewers, and the evaluation has been overseen by a Reviewing Editor and Satyajit Rath as the Senior Editor. The reviewers have opted to remain anonymous.

Essential Revisions:

1) Figure 5C shows MMF increases calcium flux in Hca2FL/FL control mice but there is no conditional knockout control. Please included data from the conditional knockout mice.

2) Please provide analysis of cells from the spleen and/or lymph nodes.

3) Address other substantive comments of the reviewers.

*Reviewer #1 (Recommendations for the authors):*

The manuscript by Kosinska, Assmann et al., investigates the effect of diet on dimethyl fumarate (DMF) treatment in experimental autoimmune encephalomyelitis (EAE). The authors found that DMF treatment led to reduced EAE clinical scores if mice were on a High Fiber diet (HFb) but not a diet with increased lauric acid (LA). Building on their prior observations that DMF acts via HCA2 on neutrophils, the authors found that the benefit of DMF treatment depended on neutrophil HCA2 expression. These findings will be valuable to the field of neuroimmunology.

The strengths of the manuscript include addressing complex interactions of DMF and diet, as well as the solid lipidomics and microbiota studies. The authors also nicely show that the diet does not alter MMF levels in vivo. The weaknesses of the manuscript include a lack of detailed immune cell analysis in EAE and small effect sizes on EAE scores in some figures.

1. Some of the EAE clinical scores are of small effect sizes that make clear interpretations difficult by examining EAE scores alone. For example, the treatment effect of DMF in NCD mice in Figure 1 appears much smaller than in the prior report (Chen et al., J Clin Invest 2014). The EAE scores in vehicle treated Hca2-/- mice in Figure 3a are lower than in WT and make it difficult to determine if there was any benefit from DMF in these mice. In Figure 6, the clinical scores and area under the curve measures look similar in both DMF treated groups. It may be helpful to expand on the point that other targets of DMF on additional cell types other than neutrophils have been previously described, such as in Linker et al., Brain 2011 and Kornberg et al., Science 2018.

2. Figure 5C shows MMF increases calcium flux in Hca2FL/FL control mice but there is no conditional knockout control. This should be included to show that MMF targets neutrophils in a HCA2-dependent manner.

3. Analysis of immunologic markers of interest in EAE are often performed on cells from the spleen, draining lymph node, or CNS tissue instead of blood. If neutrophils are the dominant HCA2-mRFP expressing cells in any of these tissues, it would help solidify that neutrophils are main cell type responsible for this effect.

*Reviewer #2 (Recommendations for the authors):*

In this manuscript, authors hypothesize that the variable response of MS patients to the drug DMF may be due to a diet effect. They show that mice subjected to EAE did not benefit from DMF treatment when fed a lauric acid-rich diet. On the other hand, mice on normal chow diet and on high-fiber (HFb) diet showed the expected protective DMF effect. DMF lacked efficacy in the LA diet-fed group despite similar resorption and preserved effects on plasma lipids. The effect observed in mice fed with the permissive HFb diet was abrogated when the hydroxycarboxylic receptor 2 (HCA2) was deleted. Furthermore, deletion of Hca2 in neutrophils was sufficient to neutralize DMF protective effects in EAE. In summary, authors showed that DMF required HCA2 on neutrophils as well as permissive dietary effects for its therapeutic action.

This is a well-organized and written manuscript, and data is presented in a clear and logical manner. Experiments are rigorously controlled, and well described. On the other hand, I have some comments and suggestions on the manuscript itself, and some concerns about the generalizability of these findings to humans.

– Use of DMT in MS vary by country. Suggest moderating the statement that DMF is the most used therapy as this may not be extensible to all places.

– EAE is typically scored using "EAE score" or "clinical score" (I've never seen it called neuro score) and it is graded in a scale of 1-5. Please adjust the Y axis to represent the entire range, so that readers can appreciate the severity. Also, the lines representing SEM are barely visible.

– Lipidomic profiles between DMF-treated and vehicle mice are more different for LAD than for HfbD. This finding seems unsupportive of the hypothesis that the effectiveness of DMF is mediated by HfbD

– Authors mention neuroprotection and neurological deficit (pages 6 and 7) although there is no evidence of that. EAE is largely an immune mediated process, and the lack of clinical activity does not imply neuroprotection, but rather failure of activated T cells to cause damage in the CNS.

– The sentence "these findings provide evidence that HCA2 mediates DMF function" is misleading, since, as authors acknowledge, this had been already reported.

– If HCA2 is required for DMF function (as shown in the null mice) and HCA is primarily expressed by neutrophils, the conclusion that "HCA2 stimulation in neutrophils is required for DMF therapeutic effects" (page 8) seems rather obvious.

– The effect of diet on neutrophils' transcriptome is interesting, but likely other cells would be impacted by diet too.

Authors describe these findings as a potential mechanism of variable therapeutic effect of DMF in humans. However, given that the data presented is exclusively in experimental models, that conclusion seems a too optimistic.

---

## [Author Response]

Essential Revisions:1) Figure 5C shows MMF increases calcium flux in Hca2FL/FL control mice but there is no conditional knockout control. Please included data from the conditional knockout mice.

We appreciate the comment and have measured intracellular calcium concentrations in neutrophils from conditional knockout mice after stimulation with MMF as suggested by the reviewers. The new data has been added as Figure 7C and D to the revised manuscript.

2) Please provide analysis of cells from the spleen and/or lymph nodes.

Following the comment of reviewer #1, we have performed FACS analysis of brain immune cells and added the new data as Figure 6 —figure supplement 2 to the revised manuscript.

3) Address other substantive comments of the reviewers.

We have addressed the reviewers’ comments.

Reviewer #1 (Recommendations for the authors):The manuscript by Kosinska, Assmann et al., investigates the effect of diet on dimethyl fumarate (DMF) treatment in experimental autoimmune encephalomyelitis (EAE). The authors found that DMF treatment led to reduced EAE clinical scores if mice were on a High Fiber diet (HFb) but not a diet with increased lauric acid (LA). Building on their prior observations that DMF acts via HCA2 on neutrophils, the authors found that the benefit of DMF treatment depended on neutrophil HCA2 expression. These findings will be valuable to the field of neuroimmunology.The strengths of the manuscript include addressing complex interactions of DMF and diet, as well as the solid lipidomics and microbiota studies. The authors also nicely show that the diet does not alter MMF levels in vivo. The weaknesses of the manuscript include a lack of detailed immune cell analysis in EAE and small effect sizes on EAE scores in some figures.

To address the perceived weaknesses of the manuscript, we have included additional data on *Hca2* expressing immune cells in the brain after EAE (see below).

1. Some of the EAE clinical scores are of small effect sizes that make clear interpretations difficult by examining EAE scores alone. For example, the treatment effect of DMF in NCD mice in Figure 1 appears much smaller than in the prior report (Chen et al., J Clin Invest 2014).

We appreciate the careful observation of the reviewer. In fact, the variation of the effect size over time was the trigger for the current study. Based on the present data, we propose that unintended changes in the formulation of the mouse diet may be responsible for some variation in the effect size in comparison to the 2014 report.

The EAE scores in vehicle treated Hca2-/- mice in Figure 3a are lower than in WT and make it difficult to determine if there was any benefit from DMF in these mice.

Notably, the scores of vehicle-treated Hca2^-/-^ mice were not significantly lower than of Hca^2+/+^ mice. We have included this information in Supplementary Table 6. Importantly, DMF-treated Hca2^-/-^ mice had significantly higher scores than DMF-treated Hca^2+/+^ mice (Figure 3B, Supplementary Table 6), demonstrating that HCA_2_ was required for the DMF effect.

In Figure 6, the clinical scores and area under the curve measures look similar in both DMF treated groups. It may be helpful to expand on the point that other targets of DMF on additional cell types other than neutrophils have been previously described, such as in Linker et al., Brain 2011 and Kornberg et al., Science 2018.

Importantly, DMF significantly reduced the clinical scores in *Hca2^Fl/Fl^* mice but had no effect in *Hca2^nKO^* animals. Nevertheless, we agree with the reviewer that HCA_2_ is not the only target of DMF treatment. In the revised manuscript, we have addressed the role of other DMF targets in additional cell types (page 13, lines 6-8). Moreover, we made sure not to imply that HCA_2_ is the only target of DMF (page 12, line 19).

2. Figure 5C shows MMF increases calcium flux in Hca2FL/FL control mice but there is no conditional knockout control. This should be included to show that MMF targets neutrophils in a HCA2-dependent manner.

We thank the reviewer for this well-taken comment. To perform these experiments in the revision, we had to breed mice with the conditional knockout. Due to some technical problems in the animal facility, this process took longer than expected and delayed the revision of the manuscript, for which we apologize. However, finally we were able to measure intracellular Ca^2+^ concentrations in MMF-treated neutrophils of *Hcar2^Fl/Fl^* and *Hcar2^nKO^* mice. The data were added to Figure 7C, D of the revised manuscript.

3. Analysis of immunologic markers of interest in EAE are often performed on cells from the spleen, draining lymph node, or CNS tissue instead of blood. If neutrophils are the dominant HCA2-mRFP expressing cells in any of these tissues, it would help solidify that neutrophils are main cell type responsible for this effect.

We thank the reviewer for this suggestion. As the CNS is the most important compartment for EAE, we compared *Hca2-mRFP* expressing cells in the brain of *Hca2-mRFP* mice on day 17 after EAE induction. Microglia, neutrophils, and monocyte-derived macrophages expressed *Hca2-mRFP*. The standardized RFP signal as a measure of *Hca2* expression per cell was not affected by diet or DMF treatment in the brain as in the blood, with the exception of microglia which contained a higher *Hca2-mRFP* signal in the DMF-treated mice on HFb diet than on LA diet. The data confirm that in addition to neutrophils microglia and monocyte-derived macrophages express *Hca2*. Whether they contribute to the effects of DMF in EAE needs to be tested in future experiments. So far, our experiments show a role for HCA_2_ in neutrophils. The new data have been added to the revised manuscript as Figure 6 —figure supplement 2.

Reviewer #2 (Recommendations for the authors):In this manuscript, authors hypothesize that the variable response of MS patients to the drug DMF may be due to a diet effect. They show that mice subjected to EAE did not benefit from DMF treatment when fed a lauric acid-rich diet. On the other hand, mice on normal chow diet and on high-fiber (HFb) diet showed the expected protective DMF effect. DMF lacked efficacy in the LA diet-fed group despite similar resorption and preserved effects on plasma lipids. The effect observed in mice fed with the permissive HFb diet was abrogated when the hydroxycarboxylic receptor 2 (HCA2) was deleted. Furthermore, deletion of Hca2 in neutrophils was sufficient to neutralize DMF protective effects in EAE. In summary, authors showed that DMF required HCA2 on neutrophils as well as permissive dietary effects for its therapeutic action.This is a well-organized and written manuscript, and data is presented in a clear and logical manner. Experiments are rigorously controlled, and well described. On the other hand, I have some comments and suggestions on the manuscript itself, and some concerns about the generalizability of these findings to humans.

We appreciate the positive evaluation and have addressed the reviewer’s concerns as outlined below.

– Use of DMT in MS vary by country. Suggest moderating the statement that DMF is the most used therapy as this may not be extensible to all places.

We agree with the reviewer and have changed the statement as follows: “One of the most commonly prescribed drugs is dimethyl fumarate (DMF)” (page 3, line 4).

– EAE is typically scored using "EAE score" or "clinical score" (I've never seen it called neuro score) and it is graded in a scale of 1-5. Please adjust the Y axis to represent the entire range, so that readers can appreciate the severity. Also, the lines representing SEM are barely visible.

We thank the reviewer for the comment and have exchanged the term “neuroscore” to “clinical score”. Due to animal welfare rules, we have to euthanize animals with a clinical score above 3.5 and used an immunization protocol that did not induce higher scores than 3. Therefore, we limited the range of the scale to 0 – 3.0. This point has been clarified in the revised manuscript (page 16, lines 4-6). As suggested by the reviewer, we have increased the thickness of the line representing SEM in Figures 1, 5 and 8.

– Lipidomic profiles between DMF-treated and vehicle mice are more different for LAD than for HfbD. This finding seems unsupportive of the hypothesis that the effectiveness of DMF is mediated by HfbD

We thank the reviewer for raising this important point. Indeed, DMF had a more pronounced effect on lipid profiles when mice were fed an LA than an HFb diet. The lipid-lowering action of DMF that it shares with the other clinically used HCA_2_ agonist, nicotinic acid, became most pronounced with a lipid-rich diet that induced some form of hyperlipidemia. However, the immune effect of DMF in EAE was independent of the lipid-lowering effect, although both effects are mediated by HCA_2_. Even when DMF did not have marked effects on the lipid profile, it protected against EAE in mice fed a HFb diet, indicating that the lipid-lowering effects are not responsible for the protection in EAE. In addition, this observation supports the drug monitoring data showing that the absorption of DMF was intact on LA diet. In response to the reviewer’s comment, we have improved the discussion of this issue in the manuscript (page 6, lines 21-24).

– Authors mention neuroprotection and neurological deficit (pages 6 and 7) although there is no evidence of that. EAE is largely an immune mediated process, and the lack of clinical activity does not imply neuroprotection, but rather failure of activated T cells to cause damage in the CNS.

We thank the reviewer for this insightful comment and have exchanged the term “neuroprotection” (page 6, line, 14; page 8, line 26).

– The sentence "these findings provide evidence that HCA2 mediates DMF function" is misleading, since, as authors acknowledge, this had been already reported.

The reviewer is right. We have revised the sentence to “These findings confirm that HCAR2 mediates the DMF action, also on HFb diet” (page 9, line 5).

– If HCA2 is required for DMF function (as shown in the null mice) and HCA is primarily expressed by neutrophils, the conclusion that "HCA2 stimulation in neutrophils is required for DMF therapeutic effects" (page 8) seems rather obvious.

Thank you very much for this comment. We understand the point and have reworded the sentence to “Interestingly, this effect was lost in *Hcar2^nKO^* mice (Figure 8A, B), demonstrating that HCAR2 stimulation of neutrophils is required for DMF therapeutic effects” (page 10, line 14). While neutrophils show the highest expression of HCA_2_, the receptor is also present in monocytes and microglia. Therefore, the experiment was necessary to prove the involvement of HCAR2 in neutrophils.

– The effect of diet on neutrophils' transcriptome is interesting, but likely other cells would be impacted by diet too.

As the current manuscript focusses on neutrophils, we limited the transcriptome study to this cell type. Although we agree with the reviewer that other cells could also be affected, we prefer to refrain from speculating about this in the manuscript.

Authors describe these findings as a potential mechanism of variable therapeutic effect of DMF in humans. However, given that the data presented is exclusively in experimental models, that conclusion seems a too optimistic.

We agree with this caveat and have added it as a final sentence (page 14, line 7).